# Contextual self-paced learning for Weakly Supervised Spatio-Temporal Video Grounding

**Akash Kumar** *
University of Central Florida
akash.kumar@ucf.edu

**Zsolt Kira**
Georgia Institute of Technology
zkira@gatech.edu

**Yogesh Singh Rawat**
University of Central Florida
yogesh@ucf.edu
Project Page: https://akash2907.github.io/cospal_webpage
Huggingface link: https://huggingface.co/akashkumar29/cospal

## Abstract

In this work, we focus on Weakly Supervised Spatio-Temporal Video Grounding (WSTVG). It is a multimodal task aimed at localizing specific subjects spatio-temporally based on textual queries without bounding box supervision. Motivated by recent advancements in multi-modal foundation models for grounding tasks, we first explore the potential of state-of-the-art object detection models for WSTVG. Despite their robust zero-shot capabilities, our adaptation reveals significant limitations, including inconsistent temporal predictions, inadequate understanding of complex queries, and challenges in adapting to difficult scenarios. We propose **CoSPaL** (Contextual Self-Paced Learning), a novel approach which is designed to overcome these limitations. CoSPaL integrates three core components: (1) *Tubelet Phrase Grounding (TPG)*, which introduces spatio-temporal prediction by linking textual queries to tubelets; (2) *Contextual Referral Grounding (CRG)*, which improves comprehension of complex queries by extracting contextual information to refine object identification over time; and (3) *Self-Paced Scene Understanding (SPS)*, a training paradigm that progressively increases task difficulty, enabling the model to adapt to complex scenarios by transitioning from coarse to fine-grained understanding. We demonstrate the effectiveness of CoSPaL on three benchmark WSTVG datasets, achieving a 3.9% absolute improvement on VidSTG and a 7.9% improvement on HCSTVG-v1.

## 1 Introduction

Spatio-temporal video grounding (STVG) is focused on identifying and localizing objects within video frames both spatially and temporally based on textual descriptions. This problem is critical for various applications, including video surveillance, autonomous driving, and general scene understanding. However, STVG presents significant challenges. Specifically, it requires not only distinguishing objects from irrelevant ones across time but also predicting the start and end timestamps of activities related to those objects. While recent works solve this problem in a fully-supervised setup (Yang et al., 2022; Jin et al., 2022; Lin et al., 2023), these approaches require extensive annotations, both temporally and spatially, which are costly and labor-intensive to acquire. Therefore, we focus on a weakly supervised setting for spatio-temporal video grounding (WSTVG), where models are trained using only video-level descriptions, eliminating the need for precise spatio-temporal annotations.

Weakly supervised learning has been studied extensively in the image domain, addressing tasks like phrase grounding (Datta et al., 2019; Wang et al., 2020; Liu et al., 2021) and referral grounding (Liu et al., 2019; 2022b), which locate objects in images based on text. Various methods have been explored, such as those leveraging coarse image-level labels or proposing complex mechanisms to

---

*Corresponding Author

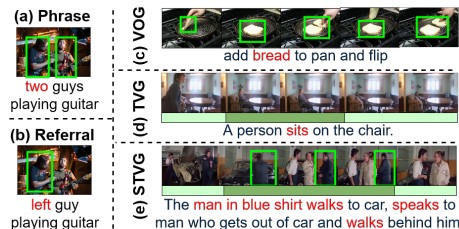

| Task | PG | RG | VOG | TVG | STVG |
|---|---|---|---|---|---|
| Video | × | × | ✓ | ✓ | ✓ |
| Referring | × | ✓ | × | × | ✓ |
| Spatial | ✓ | ✓ | ✓ | × | ✓ |
| Temporal | × | × | × | ✓ | ✓ |
| Free-form | × | × | × | × | ✓ |

Figure 1: **Comparison across tasks.** *(Left)* (a) Phrase grounding (PG) refers to grounding all nouns in the sentence, (b) Referral grounding (RG) makes the task harder by grounding specific subject, (c) Video object grounding (VOG) has fixed number of object categories and query template is fixed (d) Temporal video grounding (TVG) only focuses on temporal localization. Contrast to these, (e) STVG requires spatio-temporal grounding of specific subject using *free-form* query. Green denotes ground truth. Darker shade denotes temporal boundary. *(Right)* Table summarizes challenges involved in STVG against other tasks.

handle uncertainty in object localization. However, extending these approaches to videos adds a new layer of complexity due to dynamic changes in subject poses and scene context over time. As shown in Figure 1, STVG involves increased complexity compared to static image tasks, particularly when handling free-form textual queries, where models must understand and localize objects and actions described in natural language. Existing works that address WSTVG rely on computationally expensive solutions, such as hierarchical algorithms (Li et al., 2023) or the inclusion of extra modality data like optical flow (Chen et al., 2019b). In contrast, we propose a more streamlined and efficient approach that simplifies the process by focusing solely on visual and textual modalities.

We build upon recent progress in multimodal learning (Madan et al., 2024) and leverage vision-language foundation models as our baseline, specifically adapting Grounding DINO (G-DINO) (Liu et al., 2023), a model known for its strong zero-shot capabilities in image-level tasks. While this model shows promise for multimodal understanding, extending it to STVG reveals three key limitations (Table 1). First, it struggles with temporal consistency, frequently switching object focus across frames, as it lacks a clear understanding of temporal grounding. Second, despite being trained on large-scale image-text datasets, it finds it difficult to handle complex or imbalanced queries, particularly when multiple objects or activities are described simultaneously. Finally, the model's performance declines in dense scenes with numerous objects, where accurate localization becomes critical.

To address these challenges, we propose **CoSPaL**, a novel approach that enhances both spatial and temporal grounding in STVG. CoSPaL introduces three key components: (a) *Tubelet Phrase Grounding (TPG)*, which links textual queries to spatio-temporal *tubelets* (bounding boxes that span across frames), thereby improving object tracking over time. (b) *Contextual Referral Grounding (CRG)*, which fine-tunes the network's attention to accurately localize the relevant tubelet mentioned in the query, ensuring more precise object identification across both space and time. (c) *Self-Paced Scene Understanding (SPS)*, a training strategy that gradually increases task complexity, allowing the model to start with coarse predictions and refine them progressively. This structured approach significantly improves the model's adaptability and robustness in complex scenes.

We summarize our contributions as follows:

- We propose **CoSPaL**, the first to solve weakly supervised spatio-temporal video grounding based on a foundation model.
- We propose *Contextual Referral grounding (CRG)* which extracts contextual information from query and enhances spatio-temporal grounding ability of the network.
- We introduce *Self-paced Scene Understanding (SPS)* training scheme that makes network robust for complex challenging scenarios.

We perform our experiments on three different benchmark datasets, ViDSTG and HCSTVG-v1 and HCSTVG-v2 demonstrating effectiveness of our proposed approach. CoSPaL outperform previous state-of-the-art methods on WSTVG task by an absolute margin of 3.9% on VidSTG and 7.9% on HCSTVG-v1.

## 2 RELATED WORK

**Object Detection:** Primary research in this area involves unimodal techniques, which use a single modality. These techniques can be broadly categorized into two groups: CNN-based methods such as FasterRCNN (Ren et al., 2017) and Bottom-Up Attention (Anderson et al., 2017), and Transformer-based methods like DETR (Carion et al., 2020) and its variants (Zhu et al., 2020; Wang et al., 2022b; Liu et al., 2022a; Cai et al., 2023; Fang et al., 2022). However, unimodal detectors are trained on limited object categories, making them unsuitable for the STVG task, which involves *free-form* queries. Recently, multimodal object detection research (Li et al., 2022; Zhang et al., 2022; Yao et al., 2022; Liu et al., 2023) has emerged, taking image and text as inputs to output bounding boxes for objects. Multimodal detection involves: a) Adaptation to open-world scenarios (Minderer et al., 2022; Feng et al., 2022; Dou et al., 2022; Zhang et al., 2022; Yao et al., 2022; Li et al., 2022), allowing detection of novel objects at test time, suitable for STVG queries, and b) Strong zero-shot grounding capabilities. These foundation models (Yan et al., 2023; Wang et al., 2023; Liu et al., 2023; Cheng et al., 2024) are trained on large-scale datasets like COCO (Lin et al., 2014) and O365 (Shao et al., 2019), showing strong zero-shot performance for various tasks, including referral grounding. G-DINO (Liu et al., 2023) outperforms previous models (Yan et al., 2023) in image referral tasks. We base our work on G-DINO. ***Different*** from existing setups, we adapt G-DINO to video settings for STVG task.

**Spatio-Temporal Video Grounding:** This task involves grounding spatio-temporal tubes based on textual queries, addressing spatial and temporal dimensions. Initial solutions developed for STVG use a two-stage process with separate spatial (Rohrbach et al., 2015; Yamaguchi et al., 2017; Chen et al., 2019c) and temporal grounding (Gao et al., 2017; Chen et al., 2019a). However, pre-trained object detectors have a fixed number of object categories, limiting their effectiveness for STVG tasks with free-form queries. Recent multimodal approaches (Su et al., 2021; Yang et al., 2022; Jin et al., 2022; Lin et al., 2023; Gu et al., 2024; Wasim et al., 2024) tackle this challenge in a single stage, leveraging image-based detectors (Kamath et al., 2021), video encoders, and spatio-temporal decoders (Yang et al., 2022; Wasim et al., 2024), addressing feature alignment inconsistencies (Jin et al., 2022), or utilizing static and motion cues (Lin et al., 2023; Gu et al., 2024), . These methods typically rely on frame-level bounding box annotations for training. ***Differently*** from these, our work adopts a cost-efficient approach by refraining from using spatio-temporal labels.

**Weakly Supervised Learning** There are some existing works on dense tasks (Kumar et al., 2025; Singh et al., 2024; Kumar et al., 2023; Rana & Rawat, 2023; Kumar & Rawat, 2022; Rana & Rawat, 2022; Dave et al., 2022; Modi et al., 2022), however, they are unimodal and on semi-supervised or active learning and can't be extended to solve weakly supervised STVG task. For grounding techniques, it can be categorized into three main classes. In images, it includes phrase and referral grounding. Phrase grounding(Rohrbach et al., 2016; Datta et al., 2019; Chen et al., 2018; Akbari et al., 2019; Gupta et al., 2020; Wang et al., 2020; Liu et al., 2021; Wang et al., 2021a) highlights objects in textual queries using margin losses (Datta et al., 2019; Chen et al., 2018), contrastive optimization (Gupta et al., 2020; Wang et al., 2020), and reconstruction (Rohrbach et al., 2016) methods. Referral grounding (Liu et al., 2019; 2022b; Jin et al., 2023) adopts reconstruction (Liu et al., 2019; 2022b) or contrastive learning (Jin et al., 2023) to ground objects. In temporal grounding for videos (Wang et al., 2021b; Chen et al., 2022; Lin et al., 2020; Zheng et al., 2022a;b), both reconstruction and contrastive methods are prominent, however recent reconstruction-based approaches (Lin et al., 2020; Zheng et al., 2022a;b) outperform contrastive ones. We employ a contrastive and reconstructive approach for spatial and temporal grounding respectively. ***Different*** from existing works, we incorporate referential capabilities in spatial and temporal grounding for videos which previous work don't. Our approach induce focusing on specific contextual knowledge to enhance mutual interaction between vision and text.

## 3 METHODOLOGY

**Problem Formulation:** In WSTVG, the input is an untrimmed video $V = (v_1, v_2, ...v_L)$ of length $L$ frames, accompanied by a query description caption $Q$ describing the subject and activity in the video. The task output is the spatio-temporal tubelet for the main subject, $A_R = \{a_r\}_{t_s}^{t_e}$, where $a_r$ represents the main subject in the query, and $t_s$ and $t_e$ denote the corresponding starting and ending timestamps of the activity. In weakly-supervised settings, only video-level annotations are available for training, and there are no spatio-temporal labels for supervision.

Table 1: Comparison of weakly-supervised G-DINO(Liu et al., 2023) with previous approaches.

| Methods | VidSTG-Declarative | | | VidSTG-Interrogative | | | HCSTVG-v1 | | |
|---|---|---|---|---|---|---|---|---|---|
| | m_vIoU | vIoU@0.3 | vIoU@0.5 | m_vIoU | vIoU@0.3 | vIoU@0.5 | m_vIoU | vIoU@0.3 | vIoU@0.5 |
| AWGU (Chen et al., 2020) | 9.0 | 7.9 | 3.1 | 8.6 | 6.9 | 2.9 | 8.2 | 4.5 | 0.8 |
| Vis-CTX (Shi et al., 2019) | 9.3 | 7.3 | 3.3 | 8.7 | 7.2 | 2.9 | 9.8 | 6.8 | 1.0 |
| WINNER (Li et al., 2023) | 11.6 | 14.1 | 7.4 | 10.2 | 12.0 | 5.4 | 14.2 | 17.2 | 6.1 |
| W-GDINO (Liu et al., 2023) | 10.6 | 13.0 | 7.8 | 9.8 | 12.1 | 6.7 | 9.0 | 11.6 | 4.6 |

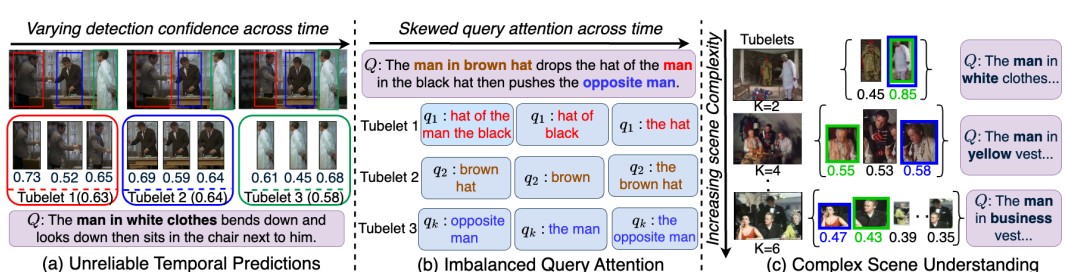

Figure 2: **Illustration of failures of W-GDINO:** (a) *Unreliable Temporal Predictions:* Foundation model predictions are inconsistent across time and switch attention between actors across time. This leads to performance degradation. (b) *Imbalanced Query Attention:* It showcases that model lacks understanding of complex queries. Across time, query which model attends to for each subject tubelet is inconsistent and doesn't match with ground truth, (c) *Complex Scene Understanding:* As the number of subjects increase, model's capability to focus on the specific subject described in query reduces. This shows it's lack of understanding of challenging scenarios. K denotes total number of subjects. Blue and red denotes predictions and green denotes ground truth in (a) and (c), and brown in (b).

## 3.1 PRELIMINARIES: GROUNDING DINO (G-DINO)

G-DINO (Liu et al., 2023) extends closed-set object detection to open-world scenarios. It takes an image and query as input, and outputs a bounding box and confidence score. In our work, we use text input query $Q$ and video frames $I_f = \{V_f\}_{f=1}^T$, with $T$ denoting the video length. As multi-modal object detectors are image-based and STVG is a video task, we first extend G-DINO for videos. To adapt it, we run detections throughout the video, storing each subject's bounding box, confidence score, and features. Applying a tracker (Aharon et al., 2022) to these detections yields *tubelets* for each detected subject $k$ as $\mathcal{T}_{o_k}$. $K$ represents the total number of subjects throughout the video. This adapted model is termed weakly-supervised Grounding DINO (W-GDINO). To assess W-GDINO's performance, we accumulate and average the confidence scores of each tubelet, selecting the one with the highest score. While Table 1 demonstrates competitive performance, we observe some issues with this approach.

We attribute these issues to three major factors: *(a) Unreliable Temporal Predictions:* Figure 2 (a) shows the model's predictions are inconsistent over time due to factors like varying subject poses and similar spatial features. W-GDINO lacks spatio-temporal localization. *(b) Imbalanced Query Attention:* GDINO is trained via byte encoding scheme which breaks down the original query and then rebuild it up. Due to this, GDINO is unable to focus on a specific part of query consistently across time, as seen in Fig. 2 (b). This causes confusion about the described subject. *(c) Limitations in Complex Scene Understanding:* WSTVG datasets present challenging scenes with many objects, as shown in Fig. 2 (c). This complicates spatial and temporal associations. We propose CoSPaL to address these limitations.

## 3.2 CONTEXTUAL SELF-PACED LEARNING (COSPAL)

CoSPaL consists of three key components to address the above limitations: Firstly, Tubelet Phrase Grounding (TPG) (Sec. 3.2.1) induces spatio-temporal localization capability in W-GDINO. It enables to remove unreliable temporal predictions by aligning textual and tubelet features for *spatial grounding* and textual and video features for *temporal grounding*. Second, to improve attention on relevant parts of query, we propose a novel concept of Contextual Referral grounding (CRG) module to extract fine-grained attributes that highlight the subject's contextual information. It enhances focus

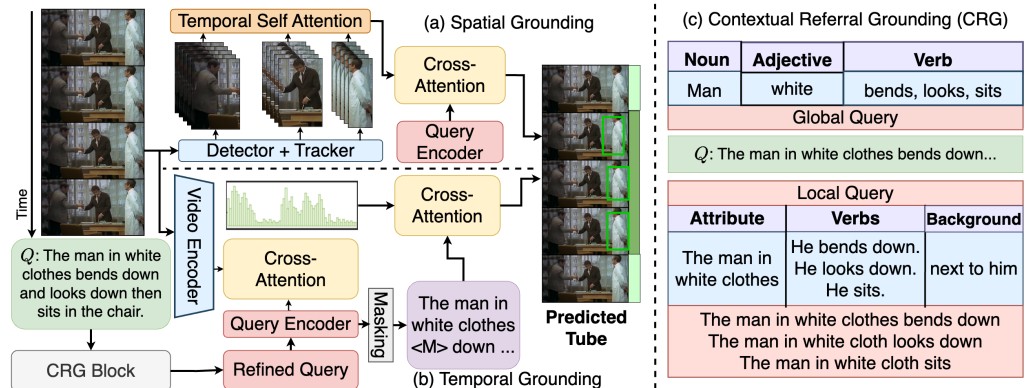

Figure 3: **Overview of CoSPaL:** TPG contains two grounding modules namely, *spatial* and *temporal*. Spatial module grounds the correct subject tubelet. Temporal module predicts the temporal action boundary via cross attention between highlighted vision features and masked query features. Contextual Referral Grounding (CRG) block shows the breakdown and generation of local ($Q_{ol}$) and global query ($Q_{og}$). Green shows predicted bounding box. Darker green shade shows predicted temporal boundary localization.

on the subject in context (Sec. 3.2.2). Finally, since STVG is challenging and matching queries with numerous scene subjects is difficult, we introduce self-paced scene understanding (SPS). It progressively increases task difficulty to adapt the network for complex scenarios and enhance the network's discriminative ability over time (Sec. 3.2.3). An overview of CoSPaL is shown in Fig. 3.

### 3.2.1 Tubelet Phrase Grounding (TPG)

TPG adapts W-GDINO to solve *spatial* and *temporal* grounding jointly. The *spatial grounding* module leverages word-level representations to enhance the alignment between textual and tubelet features. Meanwhile, the *temporal grounding* module optimizes the correspondence between video and textual features to accurately predict the start and end timestamps of the activity described in the caption. Following previous works in weakly supervised grounding (Datta et al., 2019; Gupta et al., 2020; Wang et al., 2020; 2021a) we incorporate a visual encoder to extract features from pre-trained object detectors and video encoders and a language encoder (Devlin et al., 2019; Pennington et al., 2014) to provide rich textual representations of query.

**Visual encoder:** We extract object level representations $f_{o_k} = F_o(o_k) \in \mathbb{R}^{K \times 256}$ from G-DINO (based on DETR (Carion et al., 2020)), where, $F_o$ is object encoder model, and, $o_k$ denotes $k^{th}$ detected subject. We link these detections via a tracking (Aharon et al., 2022) algorithm to generate subject tubelets for subject $k$, $\mathcal{T}_{o_k} = \{o_{k_t}\}_{t=s}^e$ where $s$ and $e$ denotes starting and ending timestamp of the subject in the video. Tubelet features for a video is represented by $\mathcal{F}_T = \{f(\mathcal{T}_{o_k})\}_{k=1}^K \in \mathbb{R}^{T \times K \times 256}$, where $K$ denotes number of objects present in a video. For video features, we utilize a video encoder, $F_v$ (e.g. I3D (Carreira & Zisserman, 2017)) to get clip-level features, $f_c = F_v(\{V_t\}_{t=1}^C) \in \mathbb{R}^{C \times 1024}$. $C$ denotes the number of clips in the video.

**Query encoder:** We pass the query $Q$ through a language encoder ($F_l$), BERT (Devlin et al., 2019), to get word level embeddings $\mathcal{F}_W = \{f_{w_m}\}_{m=1}^N \in \mathbb{R}^{N \times 768}$, where $f_w = F_l(\{q_m\}_{m=1}^N)$. $N$ denotes total words in query.

**Spatial Grounding Module** highlights the correct tubelet. We use a multimodal contrastive learning optimization to highlight the relation between words and tubelet. The insight is that to find the maximal mutual information shared between two modalities, they first need to be projected into the same space. We start with subject tubelet features in a video ($\mathcal{F}_t$). The features are extracted from DETR; thus, the features do not have any interaction amongst them temporally. To establish connection between them and enhance the features temporally, we apply a temporal self-attention block (TSA) to generate updated tubelet features, $\tilde{\mathcal{F}}_T = \text{TSA}(\mathcal{F}_T)$. This helps the network to highlight frames which provide more contextual information. For example, if the query description is "man in brown coat...", TSA give higher weights to the frame when the actor's coat is visible rather than noisy frames (frames with zoomed in faces, partial body, challenging poses, figure shown in

supplementary). We project $(\tilde{\mathcal{F}}_T)$ for each actor into a shared space by applying cross-attention block to highlight subjects mentioned in the query $(\mathcal{F}_w)$. $\tilde{\mathcal{F}}_T$ is used as key and value pairs, and, $\mathcal{F}_W$ is query. We use simple feed-forward MLP layers to project key and query features. We calculate the similarity (SIM) between individual word $f_w$ and tubelet feature $f_{\tilde{T}_k}$ as $\texttt{SIM}(f_{w_m}, f_{\tilde{T}_k}) = (\texttt{MLP}_\texttt{q}(f_{w_m})^T \texttt{MLP}_\texttt{k}(f_{\tilde{T}_k}))/\sqrt{d}$, where $\texttt{MLP}_\texttt{q}$ and $\texttt{MLP}_\texttt{k}$ denote MLP layers for key and query features. Using the features projected into the same space, we calculate aggregated attention for the video with all tubelets $T$, $\texttt{A}_\texttt{T}$ with each word as $\texttt{A}_\texttt{T}(f_T, f_{w_m}) = \sum_{k=1}^{K} \texttt{softmax}(f_{\tilde{T}_k}, f_{w_m}) \texttt{MLP}_\texttt{v}(f_{\tilde{T}_k})$, where $\texttt{softmax}$ is defined in Eq. 1.

$$\texttt{softmax}(f_{T_k}, f_{w_m}) = \frac{exp(\texttt{SIM}(f_{w_m}, f_{\tilde{T}_k}))}{\sum_{k'=1}^{K} exp(\texttt{SIM}(f_{w_m}, f_{\tilde{T}'_k}))} \tag{1}$$

$\texttt{MLP}_\texttt{v}$ is MLP layers to project value features and $\texttt{softmax}(f_{T_k}, f_{w_m})$ indicates normalized attention scores. Word features are used as query since the context in the caption is present in the scene, but the reverse may not be true. Lastly, to optimize the learning for spatial module, we apply multimodal InfoNCE loss shown in Eq. 2 to induce discriminative learning in the projection layers to pull regions with higher attention closer and push away negative tubelets farther. To get the compatibility function for loss, we update the $\texttt{A}_\texttt{T}$ as $\texttt{A}_\texttt{T} = \texttt{MLP}_\texttt{v}^T(f_{w_m})\texttt{A}_\texttt{T}$. We pick negative tubelets $(f'_T)$ within the batch.

$$\mathcal{L}_s = -\sum_{m=1}^{N} \left( \log \frac{exp(\texttt{A}_\texttt{T}(f_{T_k}, f_{w_m}))}{\sum_{k'=1, (k'\neq k)}^{K} exp(\texttt{A}_\texttt{T}(f_{T_{k'}}, f_{w_m})))} \right) \tag{2}$$

**Temporal Grounding Module** provides temporal boundary for activity mentioned in query. The limitation of the spatial grounding module is its inability to provide start and end timestamps for actions, which reduces its adaptability for the WSTVG task. We incorporate a reconstruction-based approach based on its effectiveness for temporal grounding(Lin et al., 2020; Yang et al., 2023; Zheng et al., 2022a;b). The main idea is to enforce semantic consistency between video and query. Firstly, original query highlights key segments in video. Then, original query is masked and use the highlighted visual segments features to regenerate masked query features. This enforces the video features semantically correspond to query features at test time. Fig. 3 (b) shows outline for the CRG module.

Since temporal grounding requires understanding of action, and features from the object detector contain only image-level information, we therefore acquire clip-level features $f_c$ from the video encoder model. We take cross attention (CA) between original query features $(f_q)$ to get highlighted visual features as $f'_c = \texttt{CA}(f_q, f_c)$. Key and value pairs come from the visual features and query comes from the caption. Then, the original query is passed through a masking module $\mathcal{M}$ which looks into specific part-of-speech (POS) tags of the query and mask out noun/adjectives/verb to generate the masked query, $\tilde{q}$. We use a transformer decoder (DEC) to regenerate the probability distribution of masked query as $\mathcal{P}(\tilde{q}_{w_m}|f'_c, \tilde{q}_{[0:m-1]}) = \texttt{DEC}(\texttt{CA}(f'_c, \tilde{q}))$, where, $\mathcal{P}$ denotes probability distribution for $m^{th}$ word $\tilde{q_w}$. The reconstruction loss $(\mathcal{L}_t)$ to train the model is the difference between regenerated and original query distribution shown in equation 3, where $N$ denotes total number of words.

$$\mathcal{L}_t = -\sum_{m=1}^{N} log\mathcal{P}(q_w|f'_c, \tilde{q}_{[0:m-1]}) \tag{3}$$

### 3.2.2 CONTEXTUAL REFERRAL GROUNDING (CRG)

Analyzing the original query $Q$, we observe that it contains descriptions of background objects/scene. It also contains information about attributes and actions related to those objects. In equation 2, spatial loss $\mathcal{L}_s$ applies a summation across similarity with all words. This leads to confusion for the network regarding which tubelet is actually the target tubelet (*referral subject*). CRG addresses this by leveraging referral subject's attributes to improve attention over objects sharing common information with the query. The *intuition* is that referral subject-related attributes further enhance grounding capability.

We refine this information from *free-form* text query by decomposing query $Q$ into three sub-parts: a) Referral tubelet and its attributes $(Q_{oa})$, b) Referral tubelet action verbs $(Q_{ov})$, and, c) background

information ($Q_b$). We generate new queries, $Q_o$ that describes *referral* tubelet using $Q_{oa}$ and $Q_{ov}$. This helps the network associate attributes and actions with correct tubelets ($\mathcal{T}_{o_k}$). Additionally, for a more fine-grained aspect, we look into noun-adjective-verb word features corresponding to referral from generated ($Q_o$) and original query ($Q$). These features contain relevant information in relation to the whole caption. Thus, we call these referral features as $Q_{og}$, since they contain global knowledge, and earlier query $Q_o$ as $Q_{ol}$ since they contain local knowledge in relation to the original query. Fig. 3 (c) illustrate the process of generation of $Q_{ol}$ and $Q_{og}$. The updated spatial loss ($\tilde{\mathcal{L}}_s$) is shown in equation 4, where $f_{w\langle Q_{og}:Q_{ol}\rangle}$ denotes words from updated queries.

$$\tilde{\mathcal{L}}_s = -\sum_{m=1}^{N}\left(\log\frac{exp(\mathtt{A_T}(f_{T_k}, f_{w\langle Q_{og}:Q_{ol}\rangle_m}))}{\sum_{k'=1,(k'\neq k)}^{K}exp(\mathtt{A_T}(f_{T_{k'}}, f_{w\langle Q_{og}:Q_{ol}\rangle_m})))}\right) \tag{4}$$

Similarly, for temporal localization module, existing works (Chen et al., 2022; Wang et al., 2021b; Lin et al., 2020) lacks referential capabilities. Thus, we update original query with these local queries such that attention is more concentrated on beginning and ending timestamps relevant to the referral subject. Eq. 5 shows updated reconstruction loss ($\tilde{\mathcal{L}}_t$).

$$\tilde{\mathcal{L}}_t = -\sum_{m=1}^{N}log\mathcal{P}(q_{w\langle Q_{og}:Q_{ol}\rangle}|f_c', \tilde{q}_{[0:i-1]\langle Q_{og}:Q_{ol}\rangle}) \tag{5}$$

### 3.2.3 SELF-PACED SCENE UNDERSTANDING (SPS)

STVG is inherently complex, particularly when dealing with videos lacking explicit spatio-temporal labels and containing multiple subject tubelets. The primary challenge lies in maximizing correlation between query and subject features, especially when their number increases significantly. To address this, we introduce a self-paced curriculum learning (SPL) strategy (Wang et al., 2022a; Soviany et al., 2022) to enhance optimization. This approach incrementally increases task complexity, beginning with simpler scenarios and progressively introducing more difficult ones as the model improves. By gradually exposing the model to more challenging cases, SPL ensures better convergence and robustness in learning complex spatio-temporal relationships.

SPL utilizes a student-driven difficulty scheme. Firstly, we analyze the scenes where model gets confused and then devise training accordingly. Thus, we emulate SPL in two stages: *(a) Difficulty Measure:* We measure difficult based on the scene complexity. Fig. 2 (c) shows drop in attention values on correct subject as the scene gets complex. and *(b) Training scheduler:* Based on our difficulty measure, we design the training schedule of each curriculum step by setting the upper bound on number of tubelets per video. We increase this upper bound by a factor and keep including more challenging videos with each stage and finally include all videos in last stage. This facilitates both spatial and temporal grounding module in terms of coarse-to-fine understanding of scenes. In the beginning, the network has lower discriminative power so it can understand easy (coarse) scenes better, and with time we keep increasing the difficulty and the network's ability to understand complex scenes (fine) improves.

## 4 EXPERIMENT DETAILS

**Datasets:** For our experiments, we show results on three benchmark datasets, namely VidSTG(Zhang et al., 2020), HCSTVG-v1 (Tang et al., 2020) and HCSTVG-v2 (Tang et al., 2020). VidSTG distribution comprises of 99,943 videos-sentence pairs, out of which 44,808 are declarative and 55,135 are interrogative. The total number of videos are 10,303 and it contains 80 different type of object categories. Training, validation and test contains 80,684, 8,956 and 10,303 distinct video-sentence pairs respectively and the amount of unique videos for each distribution is 5,436, 602 and 732 respectively. HCSTVG-v1 contains 4500 videos for training and 1160 videos for testing with sentence description referring to human attributes/actions. HCSTVG-v2 dataset extends version 1 to 16,544 videos. The dataset is divided into 10,131 training, 2,000 validation and 4,413 testing videos. Since test set is not available, we evaluate and show results on validation set following previous works (Yang et al., 2022; Lin et al., 2023; Gu et al., 2024).

**Implementation details:** We divide this into three parts: (a) Detection And Tracking: We utilize G-DINO(Liu et al., 2023) with 0.4 threshold for both phrase and box threshold. We run the detector

Table 2: Comparison with existing state-of-the-art methods on HCSTVG-v1 and v2 datasets.

| Methods | HCSTVG - v1 | | | | HCSTVG - v2 | | | |
| --- | --- | --- | --- | --- | --- | --- | --- | --- |
| | tIoU | m_vIoU | vIoU@0.3 | vIoU@0.5 | tIoU | m_vIoU | vIoU@0.3 | vIoU@0.5 |
| *Fully-Supervised* | | | | | | | | |
| STGVT [TCSVT20] (Tang et al., 2020) | - | 18.2 | 26.8 | 9.5 | - | - | - | - |
| STVGBert [ICCV21] (Su et al., 2021) | - | 20.4 | 29.4 | 11.3 | - | - | - | - |
| TubeDETR [CVPR22] (Yang et al., 2022) | 43.7 | 32.4 | 49.8 | 23.5 | 53.9 | 36.4 | 58.8 | 30.6 |
| STCAT [NeurIPS22] (Jin et al., 2022) | 49.4 | 35.1 | 57.7 | 30.1 | - | - | - | - |
| CSDVL [CVPR23] (Lin et al., 2023) | - | 36.9 | 62.2 | 34.8 | 58.1 | 38.7 | 65.5 | 33.8 |
| CG-STVG [CVPR24] (Gu et al., 2024) | 52.8 | 38.4 | 61.5 | 36.3 | 60.0 | 39.5 | 64.5 | 36.3 |
| VGDINO [CVPR24] (Wasim et al., 2024) | - | 38.3 | 62.5 | 36.1 | - | 39.9 | 67.1 | 34.5 |
| *Weakly-Supervised* | | | | | | | | |
| AWGU [ACMMM20] (Chen et al., 2020) | - | 8.2 | 4.5 | 0.8 | - | - | - | - |
| Vis-CTX [CVPR19] (Shi et al., 2019) | - | 9.8 | 6.8 | 1.0 | - | - | - | - |
| WINNER [CVPR23] (Li et al., 2023) | - | 14.2 | 17.2 | 6.1 | - | - | - | - |
| W-GDINO (Ours-Baseline) | 18.0 | 9.0 | 11.6 | 4.6 | 23.3 | 9.9 | 13.3 | 5.6 |
| CoSPaL (Ours) | **41.2** | **22.1** | **31.8** | **19.6** | **48.6** | **22.2** | **31.4** | **18.9** |
| | ( +23.2) | ( +7.9) | ( +14.6) | ( +13.5) | ( +25.3) | ( +12.3) | ( +18.1) | ( +13.3) |

every 5th frame and extract features from the last decoder layer. We use BoTSORT tracker(Aharon et al., 2022) algorithm to generate tubes; (b) TPG: We sample 32 frames equally indexed to get tubelet features. We extract video clip level features using I3D (Carreira & Zisserman, 2017) model. (c) CRG and SPS: We use GPT-3.5 to extract *quantifier* and *phrases* from original caption for CRG. We show more details and examples in supplementary. For SPS, we incorporate three stages of training with upper bound on four, seven and all tubelets. The model is trained for 10 epochs with 5 iterations over the dataset through each sub-phrases. More details about hyperparameters are present in supplementary.

**Inference:** We infer the subject with highest attention from spatial localization module to get the tubelet $\hat{a}$. Temporal localization module predicts the start and end temporal bounds $\hat{a}_{t_s}^{t_e}$ for the predicted tubelet.

**Evaluation Metrics:** We show performance on metrics used by previous approaches (Yang et al., 2022; Jin et al., 2022), namely mean average spatio-temporal localization (m_vIoU) and temporal localization (tIoU). vIoU and tIoU is calculated as $\frac{1}{|S_u|} \sum_{t \in S_i} IoU(\hat{b}_t, b_t)$ and $\frac{|S_i|}{|S_u|}$ respectively, where $S_i$ and $S_u$ implies intersection and union between the predicted timestamp by the model and the ground truth timestamp. $IoU(\hat{b}_t, b_t)$ calculates the spatial overlap between the predicted bounding box $\hat{b}_t$ and ground truth bounding box $b_t$ at frame $t$. m_vIoU is calculated by averaging over vIoU for all the videos in test set. vIoU@R indicates scores for samples whose mean vIoU is greater than R. We show for two values 0.3 and 0.5 following previous works(Yang et al., 2022; Li et al., 2023).

## 5 RESULTS AND ANALYSIS

*Comparison with weakly-supervised baselines:* In Tables 2 and 3, we compare our approach with previous weakly-supervised approaches. On HCSTVG-v1 dataset, we beat AWGU and Vis-CTX on all metrics by a margin of 14-15% at mean vIoU score. We outperforms the recent approach, WINNER(Li et al., 2023) by a margin of 8%. Looking closely at different IoUs, we outperform previous SOTA at 0.3 by 2x and at 0.5 by 3x. Against W-GDINO, CoSPaL outperforms it by a margin of 5.4% and 12.4% on m_vIoU and tIoU respectively. VidSTG is an extremely challenging large-scale dataset. This is also evident by the gain made by fully-supervised approaches in recent years which is less than 2%. CoSPaL outperforms previous weakly approach by 4.4% on declarative and 3.3% on interrogative settings. At higher metrics 0.3 and 0.5, our approach achieves a gain of 4-6%.

*Comparison with fully-supervised baselines:* We also compare our approach with fully supervised approaches (Tables 2 and 3). On VidSTG dataset, the proposed approach beats a few of the fully-supervised approaches which are combination of spatial and temporal grounding (Gao et al., 2017) modules. On HCSTVG-v1 dataset, we outperforms STGVT and SVGBert on all metrics. Against recent approaches(Yang et al., 2022; Jin et al., 2022; Lin et al., 2023), our approach is within 10% for

Table 3: Comparison with existing state-of-the-art methods on VidSTG dataset.

| Methods | Declarative Sentences | | | | Interrogative Sentences | | | |
|---|---|---|---|---|---|---|---|---|
| | tIoU | m_vIoU | vIoU@0.3 | vIoU@0.5 | tIoU | m_vIoU | vIoU@0.3 | vIoU@0.5 |
| *Fully-Supervised* | | | | | | | | |
| Groun-R [ECCV16] (Rohrbach et al., 2015) | - | 9.8 | 11.0 | 4.1 | - | 9.3 | 11.4 | 3.2 |
| STPR [CVPR17] (Yamaguchi et al., 2017) | 34.6 | 10.1 | 12.4 | 4.3 | 33.7 | 10.0 | 11.7 | 4.4 |
| WSSTG [ACL19] (Chen et al., 2019c) | - | 11.4 | 14.6 | 5.9 | - | 10.7 | 13.9 | 5.3 |
| STGRN [CVPR20] (Zhang et al., 2020) | 48.5 | 19.8 | 25.8 | 14.6 | 46.9 | 18.3 | 21.1 | 12.8 |
| STVGBert [ICCV21] (Su et al., 2021) | - | 24.0 | 30.9 | 18.4 | - | 22.5 | 26.0 | 16.0 |
| TubeDETR [CVPR22] (Yang et al., 2022) | 48.1 | 30.4 | 42.5 | 28.2 | 46.9 | 25.7 | 35.7 | 23.2 |
| STCAT [NeurIPS22] (Jin et al., 2022) | 50.8 | 33.1 | 46.2 | 32.6 | 49.7 | 28.2 | 39.2 | 26.6 |
| CSDVL [CVPR23] (Lin et al., 2023) | - | 33.7 | 47.2 | 32.8 | - | 28.5 | 39.9 | 26.2 |
| CG-STVG [CVPR24] (Gu et al., 2024) | 51.4 | 34.0 | 47.7 | 33.1 | 49.9 | 29.0 | 40.5 | 27.5 |
| VGDINO [CVPR24] (Wasim et al., 2024) | 52.0 | 34.7 | 48.1 | 34.0 | 50.8 | 29.9 | 41.0 | 27.6 |
| *Weakly-Supervised* | | | | | | | | |
| AWGU [ACMMM20] (Chen et al., 2020) | - | 9.0 | 7.9 | 3.1 | - | 8.6 | 6.9 | 2.9 |
| Vis-CTX [CVPR19] (Shi et al., 2019) | - | 9.3 | 7.3 | 3.3 | - | 8.7 | 7.2 | 2.9 |
| WINNER [CVPR23] (Li et al., 2023) | - | 11.6 | 14.1 | 7.4 | - | 10.2 | 12.0 | 5.4 |
| W-GDINO (Ours-Baseline) | 28.7 | 10.6 | 13.0 | 7.8 | 29.1 | 9.8 | 12.1 | 6.7 |
| CoSPaL (Ours) | **41.1** | **16.0** | **20.1** | **13.1** | **38.9** | **13.5** | **16.4** | **10.2** |
| | ( +12.4) | ( +4.4) | ( +6.0) | ( +5.3) | ( +9.8) | ( +3.3) | ( +4.3) | ( +3.5) |

Table 4: **Ablation on TPG (upper) and SPS (lower)** on different factors and stages of training. S & T denotes spatial and temporal grounding module, TSA denotes temporal attention.

Table 5: **Ablation study** on proposed sub-modules. We show the effectiveness of each module and their combinations. First row shows W-GDINO performance.

| S | TSA | T | tIoU | m_vIoU | vIoU@0.3 | vIoU@0.5 |
|---|---|---|---|---|---|---|
| ✓ | | | 26.2 | 13.5 | 17.7 | 7.3 |
| ✓ | ✓ | | 27.3 | 13.9 | 18.6 | 6.9 |
| ✓ | | ✓ | 35.2 | 18.0 | 26.3 | 14.1 |
| ✓ | ✓ | ✓ | 37.6 | 19.2 | 28.8 | 15.3 |

| Stages | m_tIoU | m_vIoU | vIoU@0.3 | vIoU@0.5 |
|---|---|---|---|---|
| I | 34.1 | 17.7 | 26.0 | 14.4 |
| II | 36.2 | 18.5 | 27.0 | 14.8 |
| III | 38.2 | 20.1 | 28.5 | 17.6 |

| TPG | CRG | SPS | tIoU | m_vIoU | vIoU@0.3 | vIoU@0.5 |
|---|---|---|---|---|---|---|
| | | | 18.0 | 9.0 | 11.6 | 4.6 |
| ✓ | | | 37.6 | 19.2 | 28.8 | 15.3 |
| | ✓ | | 35.8 | 20.2 | 30.5 | 17.6 |
| ✓ | ✓ | | 37.8 | 21.0 | 31.7 | 16.8 |
| ✓ | | ✓ | 38.2 | 20.1 | 28.5 | 17.6 |
| | ✓ | ✓ | 38.1 | 21.1 | 30.7 | 18.4 |
| ✓ | ✓ | ✓ | 41.2 | 22.1 | 31.8 | 19.6 |
| | | | ( +23.2) | ( +13.1) | ( +20.2) | ( +15.0) |

mean tIoU and within 15% at m_vIoU on both HCSTVG-v1 and v2. Fully-supervised approaches utilizes ground truth information to optimize the network, whereas our approach does not.

## 5.1 ABLATION STUDY

**Effectiveness of TPG sub-modules:** Firstly, we look into our base model, TPG. From Table 4, we observe that temporal grounding module plays a significant role. It boosts the standalone score of spatial grounding module on all metrics. mean tIoU and vIoU scores is boosted by a margin of 9% and 4.5% respectively. At 0.3 score boosts by a margin of 10% and almost 2x at vIoU@0.5. Temporal attention block improves score by 1% additionally on mean vIoU.

**Impact of SPS stages:** Table 4 demonstrates the importance of progressive learning. Increasing the difficulty with each indeed helps the network become more discriminative. We observe gains of 3% and 4% on mean vIoU and tIoU respectively.

**Effectiveness of SPS and CRG:** We analyze each sub-module in our proposed approach in Table 5. Firstly, our proposed TPG outperforms W-GDINO on all metrics. On the main metric, our method outperforms it by 10%. The context refinement grounding aspect standalone boosts the score by 11% on top of Weakly-GDINO and 1% on TPG module. This shows the impact that contextual referral matters and focus in on attributes related to *referral subject* helps. When TPG and CRG are combined, that is we utilize different referral phrases and noun-adjective-verb pairs, we observe further improvement in performance by 0.8%. Introducing SPS on TPG and CRG standalone, shows a gain of 0.9% on m_vIoU in both. This indicates that the network adapts well when the difficulty of the task is increased

The woman in blue clothes takes something on the table and wraps it around her wrists a few times.

The man in blue clothes runs to the side of the road, stops, turns around and speaks to the person in white , and then turns again.

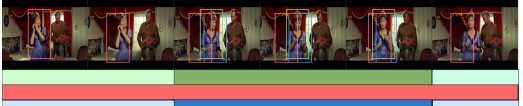
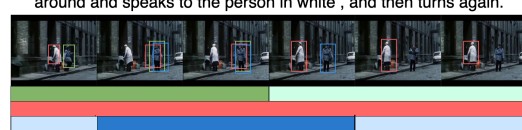

Figure 4: **Qualitative analysis:** Green: ground truth; red:W-GDINO, and blue: CoSPaL (darker shade represents temporal detection boundaries). W-GDINO suffers from temporal localization and imbalanced attention focusing on different subjects throughout the video. CoSPaL overcomes these limitations and has better overlap with GT in both scenarios.

progressively. Using both SPS and CRG with TPG performs the best (last row). It boosts the performance on top of TPS+CRG by a margin of 1.1% on mean vIoU and 3.4% on m_tIoU. Against TPG, the addition of proposed sub-modules improves the performance by 2.9% and 3.6% on m_vIoU and m_tIoU respectively. Looking specifically at higher IoU at 0.5, SPS boost the performance by a margin of 2.3, 0.8 and 2.8 on TPG, CRG, and TPG+CRG. This shows substantial evidence that SPS helps both spatial and temporal grounding module increasing its discriminative ability with task complexity.

**Impact of detector backbones:** Table 6 shows CoSPaL outperforms WINNER with Faster R-CNN (Anderson et al., 2018) backbone for fair comparison. Comparing across backbones, DETR outperforms Faster R-CNN by a margin of 6% at m_vIoU on HCSTVG-v1.

Table 6: Comparison against detector backbones.

| Methods | Detector | m_vIoU | vIoU@0.3 | vIoU@0.5 |
|---------|----------|--------|----------|----------|
| WINNER | Faster-RCNN | 11.6 | 14.1 | 7.4 |
| CoSPaL | Faster-RCNN | 16.4 | 23.7 | 11.1 |
| CoSPaL | DETR | **22.1** | **31.8** | **19.6** |

**Computational Efficiency:** Fig. 5 shows CoSPaL is computationally efficient against all fully-supervised approaches. The main reason is the use of a frozen backbone whereas fully-supervised approaches finetune the whole backbone end-to-end. Against ours, fully-supervised training time is 2-4x with 2.5x-6.5x more GPU memory requirement. We use single gpu against 8 in CSDVL(Lin et al., 2023), 16 in Tube-DETR(Yang et al., 2022) and 32 in STCAT(Jin et al., 2022) and CG-STVG(Gu et al., 2024). In terms of total memory (number of GPUs × GPU memory), our approach uses only 1-3% against fully-supervised approaches.

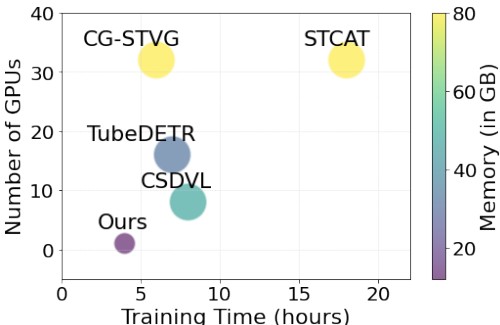

Figure 5: Comparison on computational efficiency against fully supervised approaches.

**Qualitative Analysis:** In Fig. 4, we show the effectiveness of our approach qualitatively. W-GDINO struggles with grounding the right actor as well as provides inaccurate temporal bounds, whereas our approach spatio-temporally grounds the actor better. More examples are shared in supplementary.

## 6 CONCLUSION

In this work we focus on Weakly Supervised spatio-temporal video grounding (WSTVG), aiming to localize specific objects based on textual queries without relying on labeled data. As a first step, we provide an extension of G-DINO for WSTVG task, and observe several challenges and limitations. To address these, we introduce *Contextual Self-Paced Learning* for Weakly Supervised Spatio-temporal Grounding *(CoSPaL)*. It employs *Tubelet Phrase Grounding (TPG)* module to enhance spatio-temporal prediction coherency in localization and introduces the *Contextual Referral Grounding (CRG)* module for extracting contextual information from textual queries, improving object localization precision. Additionally, the *Self-Paced Scene Understanding (SPS)* training scheme progressively increases task complexity, enhancing the network's robustness in challenging scenarios. We evaluate the proposed approach on three benchmark datasets, surpassing existing methods and demonstrating its effectiveness.

## 7 ACKNOWLEDGMENTS

This research is based upon work supported in part by the Office of the Director of National Intelligence (Intelligence Advanced Research Projects Activity) via 2022-21102100001 and the National Science Foundation under Grant No. 2239292. The views and conclusions contained herein are those of the authors and should not be interpreted as necessarily representing the official policies, either expressed or implied, of ODNI, IARPA, or the US Government. The US Government is authorized to reproduce and distribute reprints for governmental purposes notwithstanding any copyright annotation therein. The authors would like to thank Aisha Urooj Khan (Lunit) and Rohit Gupta (UCF CRCV) for helpful discussions on weakly supervised grounding and multimodal aspects.

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

## A   APPENDIX

Here, we provide some more details about our approach along with additional results and visual analysis. We also include tables which we were not able to include in main paper due to space limitations.

- Section B: We address the challenges and limitations of detector and tracker.
- Section C: Qualitative Analysis on the model's predictions.
- Section D: We show more discussion and analysis.
- Section E: Training details about architectures, datasets, and, other hyperparameters.
- Section F: Qualitative Analysis on Detection and tracking, success and failure cases and analysis on the video in the wild.

## B   CHALLENGES AND LIMITATIONS

STVG datasets are extremely challenging, especially the HCSTVG-v1 and HCSTVG-v2 where even detection and tracking fails, shown by maximum upper bound achievable in Table 12b. The HCSTVG datasets contains sudden zoom shots, scene changes, and defocus, where even good detectors fail. The additional pre-processing to track the detections to generate tubelets introduce more noise and struggles to track the right person with person crossover, scene change (very high displacement in bbox leads it to assign different IDs), view change and only partial body availability. Due to these two main limitations, we propose to solve the task by breaking it into two sub-tasks. A future work involves exploiting temporal modeling associated with each individual object jointly; however, in our current approach, we show promising results quantitatively and qualitatively.

## C   QUALITATIVE ANALYSIS (MAIN ARCHITECTURE)

In Fig. 6, we show the effectiveness of our approach qualitatively. W-GDINO struggles with grounding the right actor as well as no temporal bounds, whereas our approach spatio-temporally grounds the actor better than baseline.

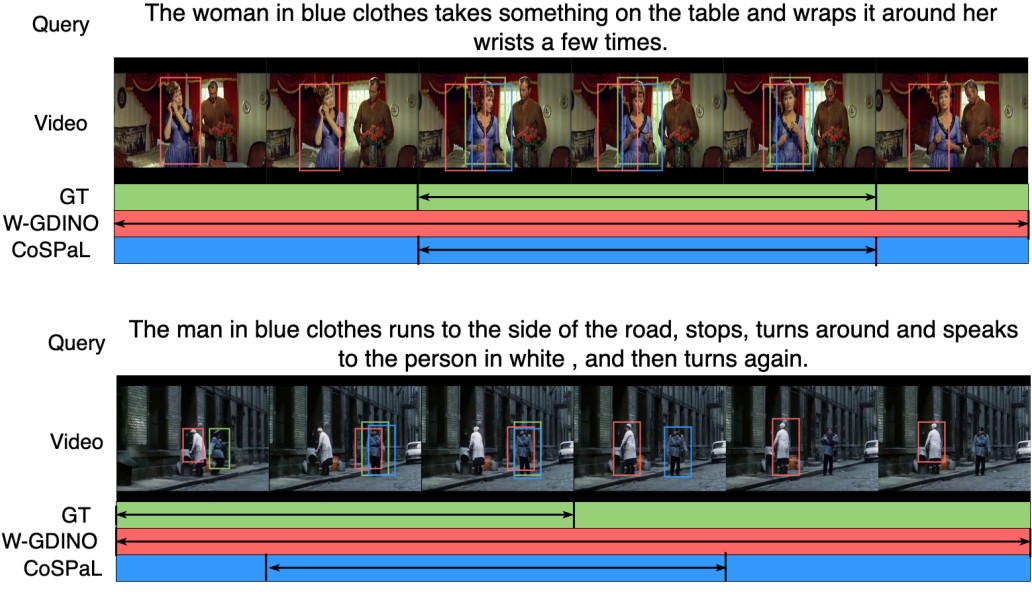

Figure 6: **Qualitative analysis:** We observe that W-GDINO detects without considering the context of the query, which is improved using the proposed method.

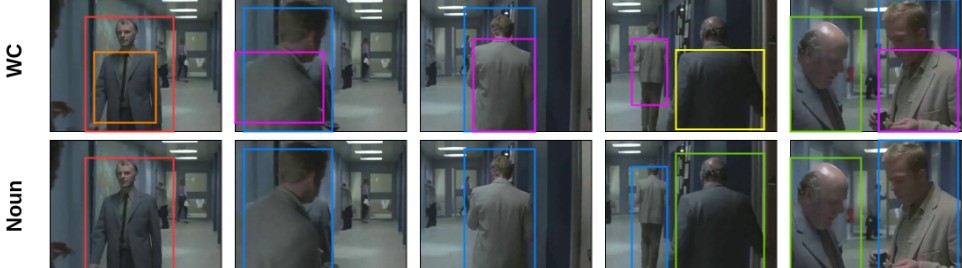

Figure 7: **Comparison between GDINO query: Whole caption (WC) vs Noun.** The first row shows detection boxes for whole query as input to the GDINO against noun extracted from the query in second row. We observe that it focuses on other objects (for eg. suit (shown in orange, pink, yellow)) which may not be the target instance but overlapping with target instance and thus helps in better score. (Tab 7). Query for the above video (WC): `The bald man leaves the room pulls the door walks towards the man in the white suit and then turns to face the white suit man.` Noun: `'man'`.

## D  DISCUSSIONS

We include multiple discussions to support and strengthen the claims in our main paper:

**Performance with Whole Caption (GDINO Input):**   In the main paper, we follow the traditional weakly supervised settings for *fair comparison* with previous SOTA, where at train and test time the detector outputs *ALL* human/object bounding boxes, and, given the query, the output should be object tubelet with maximum attention. In another setting, we analyze sending in the original caption and perform tracking on output detections. We have shown the difference in detection with only sending noun vs whole caption in Fig. 7. WC setting output detections which doesn't correspond to all subjects or overlapping detections to specific subject. In Table 7 we compare three settings. Training and testing on noun extracted from query (Noun), Train and test with whole caption (WC), and, finally, Train on WC and test with Noun. Looking at second row, input to Grounding DINO with extra information helps. To compare it with traditional weakly settings, third row we perform test with detections using Noun output. This study suggests that whole captions as query generates better detections Grounding DINO, although it might not adhere to traditional weakly-supervised settings.

Table 7: Grounding DINO Input: Noun vs Whole Caption.

| Train | Test | tIoU | m_vIoU | vIoU@0.3 | vIoU@0.5 |
|-------|------|------|--------|----------|----------|
| Noun  | Noun | 37.6 | 19.2   | 28.8     | 15.3     |
| WC    | WC   | 34.5 | 22.7   | 32.5     | 18.2     |
| WC    | Noun | 35.0 | 18.6   | 26.8     | 15.0     |

**Improvement in performance with SPS:**   From Table 5, we consistently observe a 2-3% boost for each setting with inclusion of SPS. This shows that increasing scene understanding is complementary to both baseline and baseline+CRG settings. Going in-depth analysis, in Tables 8a - 8c, we show the improvement by SPS based training for all three settings - TPG only, CRG only, and, TPG + CRG. Self-paced learning boosts score in each of the settings by 2.4, 3.4, and, 5.0 respectively. This shows the efficacy how self-paced scene understanding training paradigm helps network become more discriminative with time both spatially and temporally. This is also corroborated by the fact that training via SPS paradigm outperforms single-stage training on the whole dataset (shared in Table 5 main paper).

**Analysis on Text encoder:**   Grounding DINO finetunes the vision encoder but keeps the text encoder fixed. The vision backbone is fixed to Swin-T. For textual features, we explore two choices to find the best alignment between vision and text to begin with. From Table 9, BERT outperforms CLIP on the baseline settings, TPG. Thus, we choose BERT as encoder for all our experiments.

Table 8: Analysis on SPS in all three situations.

(a) TPG only.

| Stages | m_tIoU | m_vIoU | v@0.3 | v@0.5 |
|--------|--------|--------|-------|-------|
| I | 34.1 | 17.7 | 26.0 | 14.4 |
| II | 36.2 | 18.5 | 27.0 | 14.8 |
| III | 38.2 | 20.1 | 28.5 | 17.6 |

(b) CRG only.

| Stages | tIoU | m_vIoU | v@0.3 | v@0.5 |
|--------|------|--------|-------|-------|
| I | 33.4 | 17.7 | 24.6 | 14.8 |
| II | 36.3 | 19.6 | 28.8 | 16.3 |
| III | 38.1 | 21.1 | 30.7 | 18.4 |

(c) TPS and CRG.

| Stages | tIoU | m_vIoU | v@0.3 | v@0.5 |
|--------|------|--------|-------|-------|
| I | 32.3 | 17.1 | 24.4 | 14.0 |
| II | 37.2 | 19.9 | 28.9 | 16.7 |
| III | 41.2 | 22.1 | 31.8 | 19.6 |

Table 9: Choice of Textual Encoder: CLIP vs BERT.

| Encoder | tIoU | m_vIoU | vIoU@0.3 | vIoU@0.5 |
|---------|------|--------|----------|----------|
| CLIP | 35.7 | 18.8 | 28.8 | 14.8 |
| BERT | 37.6 | 19.2 | 28.8 | 15.3 |

**Study on Decoder Layer features:** We perform an analysis on TPG with different decoder layer features. Since G-DINO shares architecture with DETR, we extract features from six layers of decoder and ran our baseline. In Table 10, we show the performance with features from different decoder layers. We observe features from decoder layer 1 performed the best. To further refine background noise, we restrict the number of tubelets for our settings to 10. The last row (Table 10) shows that it further boost the performance by 0.8%.

**Standalone classification and temporal scores:** We perform standalone analysis on classification accuracy and temporal grounding metrics from previous works (Zheng et al., 2022a;b; Lin et al., 2020) in Table 11. In classification accuracy, we observe our approach outperforms W-GDINO by 20% and baseline TPG by 3.2%. For temporal IoU metrics, we observe including contextual phrases boost the performance further at all IoUs.

**Analysis on multiple IoUs:** In Table 12a, we show performance comparison ranging from 0.1 till 0.7 on HCSTVG dataset. CoSPaL outperforms TPG and W-GDINO at all IoUs. Our proposed approach is more effective at higher IoUs, showing a gain of 4.3% and 4.1% at 0.5 and 0.7 IoU respectively. We perform similar analysis on VidSTG dataset comparing performance at multiple IoU ranging from 0.1 till 0.7. Tables 13a and 13b shows that proposed approach outperforms both W-GDINO and TPG at all IoUs.

**Upper bound Analysis:** To quantify how challenging HCSTVG-v1, HCSTVG-v2 and VidSTG datasets are, we perform an analysis to find the upper bound, that is maximum achievable results. This analysis is necessary since it tells how challenging detection and tracking is on these datasets. We set the temporal bound 100% from ground truth. Looking at Table 12b, if the network works perfectly, our proposed module can achieve max 62.3, 52.5, 45.3, 39.8 m_vIoU on HCSTVG-v1, HCSTVG-v2, VidSTG-Declarative, and, VidSTG-Interrogative respectively. With respect to that our current approach achieves effective performance of 35.4, 42.3, 28.5, 28.6 percentage of maximum achievable.

# E EXPERIMENT DETAILS

## E.1 DETECTION AND TRACKING

**Detector:** Grounding DINO involves two hyperparameters namely text and box threshold. We set it to 0.4 for both. Setting a lower or higher values leads to oversampling or missed detections. Since dataset contains multiple resolution of images, we set the image width to 480 if original frame width is less than 550, else 800.

**Tracker:** The parameters set for BoTSORT tracker are: 1) new track threshold: 0.21, 2) Low track threshold: 0.1, 3) High track threshold: 0.34, 4) Matching threshold: 0.21, 5) Appearance threshold: 0.48, and, 6) Buffer frames: 60 to keep track of the object id for 60 number of frames.

Table 10: Comparison with different decoder layer features. Last row † shows further refinement to restrict upper bound on number of tubelets help.

| Layer | m_tIoU | m_vIoU | vIoU@0.3 | vIoU@0.5 |
|-------|--------|--------|----------|----------|
| I | **35.8** | **18.4** | 26.7 | **15.3** |
| II | 35.4 | 18.0 | **26.9** | 15.0 |
| III | 35.6 | 17.7 | 25.7 | 14.3 |
| IV | 34.4 | 17.8 | 26.2 | 14.9 |
| V | 33.5 | 18.1 | 26.4 | 14.9 |
| VI | 34.6 | 17.9 | 26.1 | 15.2 |
| I † | 37.6 | 19.2 | 28.8 | 15.3 |

Table 11: Analysis on standalone classification accuracy and temporal IoU.

(a) Classification Accuracy.

| Method | Acc. |
|--------|------|
| W-GDINO | 18.7 |
| TPG | 35.5 |
| CoSPaL | 38.7 |

(b) Temporal IoU.

| TPG(Query) | NAV(Phrases) | IoU@0.1 | IoU@0.3 | IoU@0.5 |
|------------|--------------|---------|---------|---------|
| ✓ | | 74.1 | 54.1 | 23.0 |
| ✓ | ✓ | 76.2 | 55.6 | 23.8 |

Table 12: Analysis on multiple factors showcasing effective of our proposed approach. In Table 12b, VidSTG-D means VidSTG Declarative and VidSTG-I means VidSTG Interrogative.

(a) Analysis on multiple IoUs on HCSTVG dataset.

| Method | m_vIoU | v@0.1 | v@0.2 | v@0.3 | v@0.5 | v@0.7 |
|--------|--------|-------|-------|-------|-------|-------|
| W-GDINO | 9.0 | 25.9 | 17.3 | 11.6 | 4.6 | 0.7 |
| TPG | 19.2 | 43.1 | 36.2 | 28.8 | 15.3 | 5.4 |
| CoSPaL | 22.1 | 45.6 | 38.7 | 31.6 | 19.6 | 9.5 |

(b) Upper-bound Analysis.

| Dataset | m_tIoU | m_vIoU | vIoU@0.5 |
|---------|--------|--------|----------|
| HCSTVG-v1 | 79.2 | 62.3 | 69.5 |
| HCSTVG-v2 | 76.3 | 52.5 | 54.6 |
| VidSTG-D | 66.9 | 45.3 | 46.8 |
| VidSTG-I | 66.2 | 39.8 | 39.2 |

Table 13: Analysis on multiple IoUs showcasing effectiveness of our proposed approach.

(a) VidSTG-Declarative.

| Method | m_vIoU | v@0.1 | v@0.2 | v@0.3 | v@0.5 | v@0.7 |
|--------|--------|-------|-------|-------|-------|-------|
| W-GDINO | 10.6 | 25.0 | 17.6 | 13.0 | 7.8 | 4.1 |
| TPG | 12.9 | 28.2 | 20.9 | 16.2 | 9.9 | 5.6 |
| CoSPaL | 16.0 | 33.6 | 25.8 | 20.1 | 13.1 | 7.8 |

(b) VidSTG-Interrogative.

| Method | m_vIoU | v@0.1 | v@0.2 | v@0.3 | v@0.5 | v@0.7 |
|--------|--------|-------|-------|-------|-------|-------|
| W-GDINO | 9.8 | 23.2 | 16.5 | 12.2 | 6.7 | 3.5 |
| TPG | 11.4 | 26.8 | 18.8 | 14.0 | 8.0 | 4.5 |
| CoSPaL | 13.5 | 30.3 | 22.0 | 16.4 | 10.2 | 5.7 |

### E.2 ARCHITECTURE HYPERPARAMS SETTINGS

**Weakly-GDINO:** For weakly-GDINO, we input whole text as the query and frame from video as image input. Frames are sample with a stride of 5. To calculate the GDINO predictions for a video, Firstly, we run the tracker to generate all tubelets in the video. To evaluate, we average the confidence of each tubelet across temporal dimension. The predicted tubelet is assigned to the the tubelet with highest average confidence score. The starting and ending timestamp of the predicted tubelet is used for temporal IoU calculation.

**Tubelet Phrase Grounding:** It contains two modules - spatial and temporal grounding. The batch size is set to 32. In spatial grounding module, we use Adam optimizer with a learning rate of 1e-4. The maximum length for number of words in text is set to 25 for HCSTVG. Temporal grounding module had Adam optimizer with learning rate 4e-4.

**Contextual Referral Grounding** We use GPT-3.5 to extract referral tubelet attributes ($Q_{oa}$) and referral tubelet action verbs ($Q_{ov}$). The input query $Q_a$ and $Q_v$ to the GPT to extract $Q_{oa}$ and $Q_{ov}$ respectively as below:

$Q_a$: Extract the quantifier phrase describing the main
person.
$Q_v$: Break the complex actions into simpler actions.

We provide few examples of original texts and extraction from GPT-3 for both scenarios. For first
case, extraction of main obejct in context and attributes related to its are as follows:

$Q1$: The bearded woman walks to the woman in gray
clothes and touches her face.
$A1$: The bearded women.
$Q2$: The man in the brown hat drops the hat of the
man in the black hat then pushes the opposite man then
turns and punches the man in the back.
$A2$: The man in the brown hat.
$Q3$: The woman with yellow hair walks from the right
to the left of the man in leather then pulls his arm
away.
$A3$: The woman with yellow hair.

In case of main actor and it's attribute extraction, GPT-3 worked perfectly. However, breaking
complex actions into sub-actions, GPT-3 faced challenges and sometimes hallucinates which activity
belongs to which actor. One *success* case as follows:

$Q1$: The bald man leaves the room pulls the door walks
towards the man in the white suit and then turns to
face the white suit man.
$P1$: The bald man leaves the room.
$P2$: He walks towards the man in white suit.
$P3$: He turns to face the white suit man.

One *failure* case as follows:

$Q1$: The man in the black military uniform catches
the things thrown by the opposite man with both hands
turns and bends over to pick up his hat and puts on
it.
$P1$: The man in the black military uniform catches the
things.
$P2$: He throws the thing.
$P3$: He turns and bends over.
$P4$: He pick up his hat.

In above scenario, P2 relates to the activity by the actor not in main context. We filter out these
phrases by looking into verbs in active tense. Those verbs denote activity performed by the main
actor.

**Self-paced Scene understanding:** In SPS curriculum based learning, we set the upper bound
on the number of object tubelets per video. The first stage bound is set to videos with only upto 4
tubelets and it's incremented by 3 in each stage for two more stages. In last stage, the number of
tubelets is 10 and it contains all the videos.

### E.3 COMPUTE REQUIREMENTS

For our work, we run our models on single 16 GB Tesla V100 GPU with a batch size of 32. The
training time for HCSTVG-v1 is 4-5 hours, HCSTVG-v2 id 7-8 hours and VidSTG it's 10-12 hours.

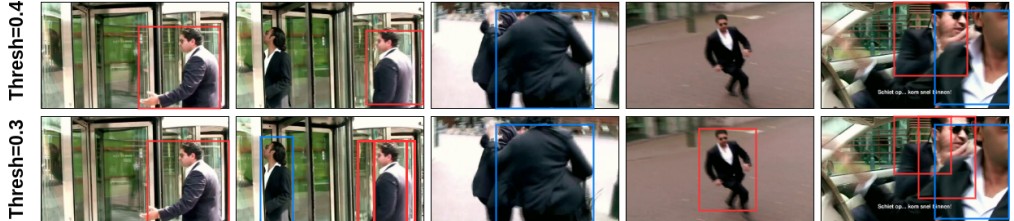

Figure 8: **Comparison between threshold for GDINO:** The first row shows detection boxes with threshold set to 0.4 and the second row shows the detection with threshold set to 0.3. We see few missed detections in earlier case, however, in later, overlapping detection issues arises. Even in second scenario, in third frame lowering confidence didn't help. The detection was missed. Query text: Noun: `'man'`.

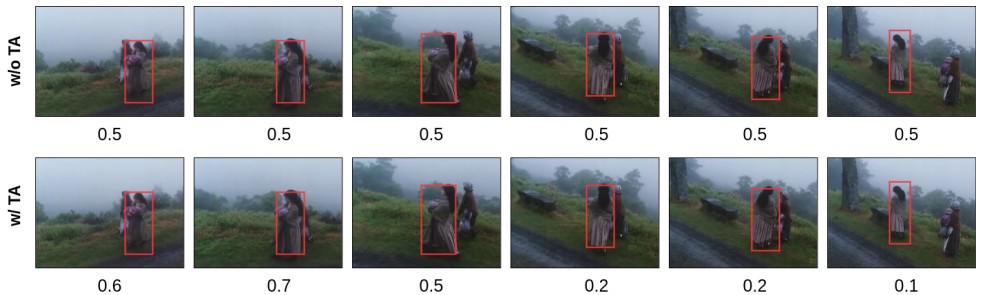

Figure 9: **Effect of Temporal Attention:** Without temporal attention (w/o TA) in first row, we observe that each frame gets equal weight, however, utilizing temporal attention (w/ TA, second row) increases weight on key frames and decrease weight for non-important frames in relation to query. `Query:` `The` **`woman holding the child`** `walks to the side of a stone` `bench stops hands the child to the woman next to her and walks to` `the front of the stone bench`

### E.4 SOCIETAL IMPACT

The proposed work could be used for surveillance and if the query is not descriptive enough can ground the wrong person leading to possible harm. However, on the positive aspect, the proposed work is free of biasness issues due to use of foundation models (trained on bigger datasets) and can be deployed in wild.

## F QUALITATIVE ANALYSIS

### F.1 FAILURES IN DETECTION AND TRACKING

In this qualitative analysis, we show the inherent failure of Grounding DINO(Liu et al., 2023) and tracker (Aharon et al., 2022).

### F.1.1 DETECTION FAILURE

In Fig. 8 we show that GDINO fails to detect the person. If we reduce threshold, it is able to detect, but, then it leads to overlapping detections which will add one another step of post-processing of non-maxima suppression.

### F.1.2 TRACKING FAILURE

There are two type of failure that happens in tracking: 1) Assigning same ID to different objects, and, 2) Different IDs to same objects. In both scenarios, tubelet features get impacted. Fig. 13 illustrates both the failures.

Query  The man in black shirt goes towards the man in brown coat and picks up the book.

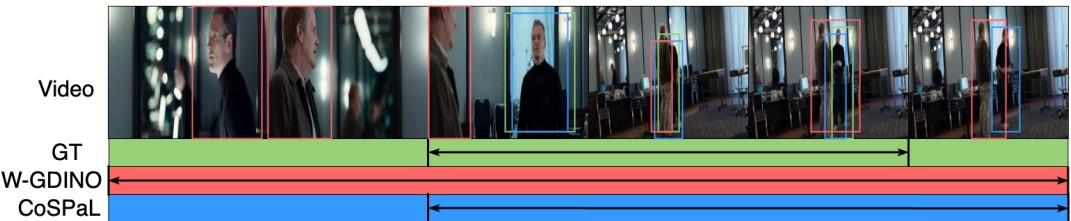

Figure 10: **Qualitative Analysis:** W-GDINO struggles to attend to the query and switch between actors across time. Our proposed approach is able to detect the main actor in context (from textual query) almost correctly spatio-temporally.

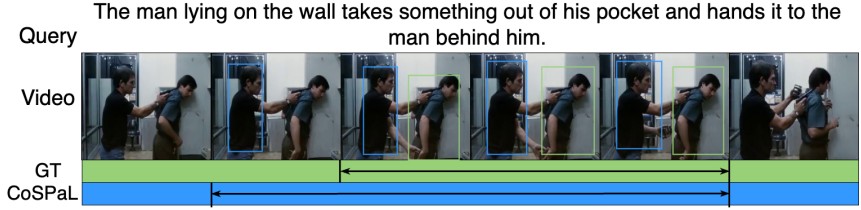

Figure 11: **Qualitative Analysis (*Failure scenario*):** In these scenarios, visual features are quite similar and query description is challenging to extract the attributes related to the main actor in context.

### F.2    Effect of Temporal Attention

In this analysis we show how temporal attention applied over tubelet helps. Fig 9 shows impact of with and without temporal attention. With temporal attention across temporal dimension, key frames that has higher mutual information in relation to query is given higher weight.

### F.3    Random Video Analysis - In the Wild

We take a random video from the internet and run our proposed approach. In Fig. 10, we show the comparison between ours against W-GDINO. We pick a video from a movie scene Steve Jobs and ran our detector and tracker and then use trained weights to predict the tubelet given the query. We formulate the query and video length on our own for this experiment.

### F.4    Success and Failure cases

Fig. 11 shows a failure scenario of our model. We observe model fails when query description doesn't explicitly contains specific attributes describing the main actor in context and spatial features of objects are very similar.

Fig. 12 shows a success scenario. In first example (*top row*), since the model doesn't contains any information about background or other actors, W-GDINO in this scenario works. However, since it doesn't have understanding of time, our approach is temporally localize the action. *Bottom row* shows a challenging example where our method performs better. In general, proposed approach works good when the query contains attributes related to main actor (referral). This shows that our proposed use of Contextual Referral grounding aspect helps in the scenario.

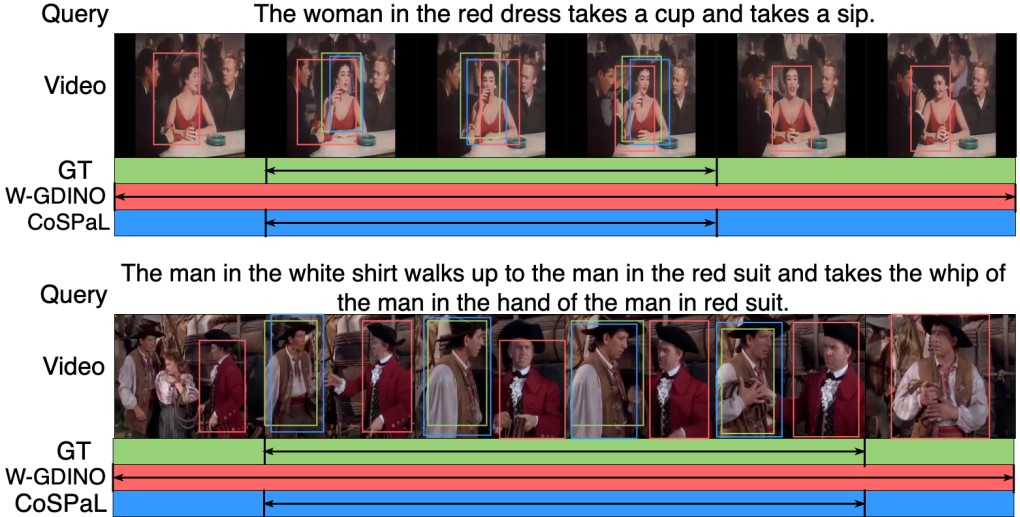

Figure 12: **Qualitative Analysis (*Success scenario*):** The proposed approach is able to properly spatio-temporally localize the actor and activity associated with it. *Top Row:* shows an easy example where W-GDINO also succeeds since the query contains description about one actor. However, it lacks temporal understanding and thus unable to localize the activity temporally. *Bottom row:* It shows a very hard example where there are query contains description about multiple actors in context. W-GDINO focuses on the background actor whereas our work is able to properly spatio-temporally localize the correct tubelet (referral tubelet).

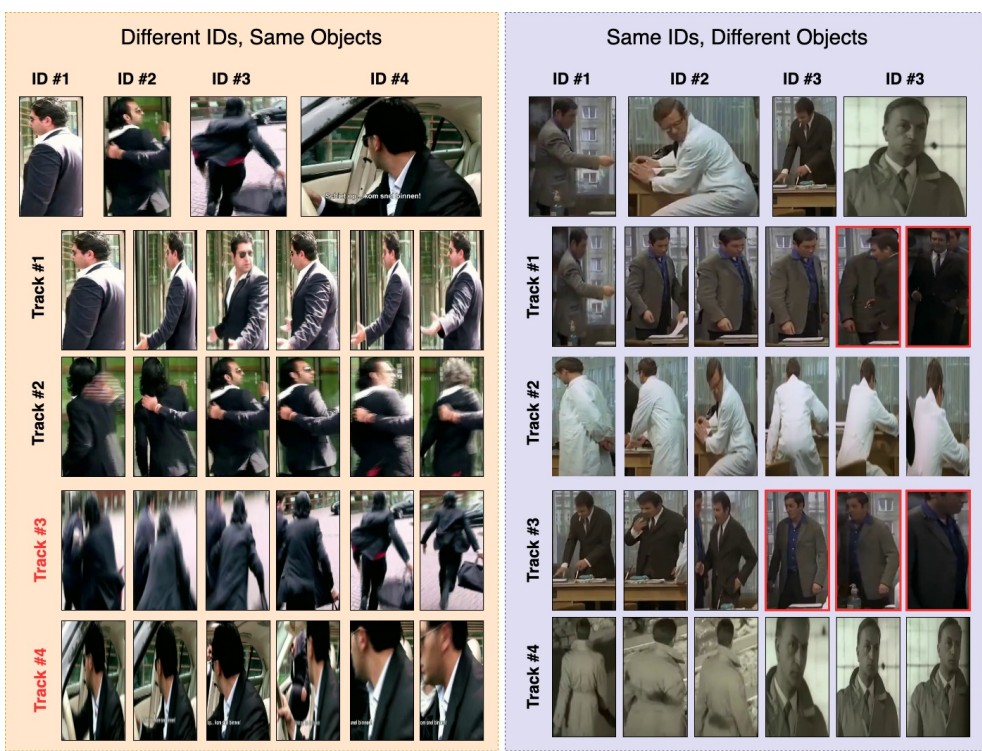

Figure 13: **Tracking failures:** *Left:* Different IDs, Same Objects - Tracks in red color are repetition of same earlier ID but assigned a new track. Tracks 1 and 4 are same IDs, and, tracks 2 and 3 are same IDs, but assigned different track IDs; *Right:* Same IDs, Different Objects - red boxes denotes switching of ID happened. Same id is assigned even if the object/actor is different.

