# OpenReview forum: "Contextual Self-paced Learning for Weakly Supervised Spatio-Temporal Video Grounding"
_ICLR.cc/2025/Conference — ICLR 2025 Poster_

### Official Review · Reviewer_tfEJ · 2024-10-31

**Soundness:** 1
**Presentation:** 2
**Contribution:** 2
**Rating:** 3
**Confidence:** 4

**Summary:**

This paper targets the Weakly Supervised Spatio-Temporal Video Grounding (WSTVG) task, which tries to retrieve specific objects and segment by sentence queries without relying on labeled data. The spatial and temporal grounding modules are used to mine the spatio-temporal information. Also, a contextual referral grounding module is utilized to extracts contextual information from query. Finally, a self-paced curriculum learning strategy is adopted for optimization. Some experiments are conducted.

**Strengths:**

Three WSTVG benchmarks are used for performance evaluation.

The compared methods are state-of-the-art. CG-STVG and VGDINO are published in 2024.

The figures and tables are clear.

Various figures are used to present the performance.

**Weaknesses:**

In Abstract, authors state that “we first explore the potential of state-of-the-art object detection models for WSTVG”. In fact, it is wrong since many STVG methods use object detection models, e.g., [1].

What is the difference between “self-paced scene understanding” and “self-paced curriculum learning”. In Section 3.2.3, authors use “self-paced scene understanding” as the section title, but all the contents describe the self-paced curriculum learning strategy. Besides, the strategy has been used in many works (Wang et al., 2022a; Soviany et al., 2022). Why you treat it as your third contribution and your title?

The technological novelty is not enough for ICLR. Most modules in the presented network in Figure 3 are popularly used. For example, cross attention in spatial grounding and temporal grounding.

Authors should conduct the main abalation study in the main paper.

The start and end timestamps need to be added in Fig 4.

Some grammatical errors. For example, "three benchmark WSTVG datasets" in Abstract.

[1] Weakly-Supervised Spatio-Temporal Video Grounding with Variational Cross-Modal Alignment

**Questions:**

Please address the weakness.

---

> ### Author Response · Authors · 2024-11-21
>
> We sincerely thank the reviewer for the valuable feedback and analysis of the paper. We have addressed the questions and concerns raised by the reviewer.
>
> **W.1 Clarification on object detection method with [1].**
>
> There's a little misunderstanding here. [1] utilizes a Faster RCNN backbone whereas we use GroundingDINO as our object detector approach. Comparing the performance of both approaches on COCO, object detection dataset (GroundingDINO [ECCV'24] Table 2 main paper), Grounding DINO outperforms Faster RCNN by a margin of 15%. We share the performance in Table *R41*. All previous works are based on Faster RCNN. For this reason, we mentioned *"first to explore the potential of state-of-the-art object detection models for WSTVG”*. We will revise the manuscript with the text - *“first to explore the potential of multimodal foundation models for WSTVG”*.
> | Method | f-mAP@0.5 |
> |-|-|
> | Faster RCNN | 42.0 |
> | Grounding DINO | 56.9 |
>
> *Table R41: Performance on COCO dataset.*
>
> However, we have shown that our good performance is simply not because of using a better object detector. In Table 2 & 3 main paper, we show our final approach outperforms the Grounding DINO baseline by a margin of 13.1%, 12.3%, 5.4%, and 3.7% at mean vIoU on HCSTVG-v1, v2 VidSTG Declarative and Interrogative respectively.  Additionally, we show the performance of our approach against the Faster RCNN backbone on HCSTVG-v1 in Table 6 main paper.  We show a comparison with [1] and [2], a recent weakly supervised approach in Table *R42*.
>
> | Method | Detector |  mvIoU | vIoU@0.3 | vIoU@0.5 |
> |-|-|-|-|-|
> |WINNER[2] | Faster RCNN | 11.6 | 14.1 | 7.4 |
> |VCMA[1] | Faster RCNN | 14.6 | 18.6 | 5.8|
> | CoSPAL| Faster RCNN | 16.4 | 23.7| 11.1|
> | CoSPAL | Grounding DINO | 22.1 | 31.8 | 19.6 |
>
> *Table R42: Comparison against detector backbones on HCSTVG-v1 dataset.*
>
> CoSPAL, our approach outperforms VCMA[1] by a margin of 1.8%, 5.1%, and 5.3% at mean vIoU, vIoU@0.3, and vIoU@0.5 respectively with similar backbone. Utilizing Grounding DINO features, CoSPAL outperforms VCMA[1], 7.5%, 13.2%, and 13.8% at mean vIoU, vIoU@0.3, and vIoU@0.5. This shows the efficacy and robustness of our approach with different detector backbones. We will update the manuscript to update these numbers in the main paper.
>
> **W.2 Difference between Self-paced scene understanding and Self-paced Curriculum Learning.**
>
> Self-paced Curriculum Learning is a branch of curriculum learning that involves designing an easy-to-hard training scheme based on a difficulty measure. We agree with the reviewer that it's a general concept and has been explored in several domains such as image classification, object detection, segmentation, etc. To the best of our knowledge, our work is the first to study curriculum-based strategy for the WSTVG task. The main challenge is to assess *Difficulty measurer*. Designing scene/sample difficulty for the WSTVG task is our contribution. Previous works mentioned in [Wang et al. 2022a, Soviany et al. 2022] explored it for trivial tasks which are not directly applicable or can be extended to dense video tasks such as ours, mostly for one significant reason: Difficulty is defined on the basis of whether images/regions are difficult to classify from a fixed set of classes whereas STVG task is a free-form textual query understanding. The existing approaches can't deduce a notion of scene difficulty for the WSTVG task.
>
> Self-paced scene understanding deduces the difficulty measure based on spatio-temporal scene complexity. We show qualitatively in Fig 2 c main paper limitations of the baseline model. Quantitatively, analyzing TPG (adaptation of foundation model), we analyze the scenes based on spatio-temporal complexity and observe the maximum failure rate of baseline adaptation, TPG. We propose an empirical approach to determine schedule stages suitable for the WSTVG task, and we also demonstrate that this is generalizable to other datasets.
>
>
> **W.3 Technical Novelty (Fig. 3).**
>
> Fig. 3 provides an overview of the main modules utilized for the adaptation of image-based foundation models for the WSTVG task. Our contribution is an effective adaptation of multimodal foundation models for this task, which to the best of our knowledge has not been explored earlier. We agree with the reviewer that cross-attention or contrastive learning are existing components, but we merely use these to develop a strong baseline since a trivial adaptation was not found effective (shown in Table 1 main paper). These existing components are carefully utilized considering the limitations of the trivial extension. The proposed method CoSPaL uses this adaptation as a baseline.
>
> CoSPaL consists of novel components: 1) Contextual referral Grounding (CRG) and 2) Self-paced Scene understanding (SPS) modules which to the best of our knowledge have not been explored earlier.

---

> > ### Comment · Reviewer_tfEJ · 2024-12-03
> > **Keep the rating due to the weak novelty**
> >
> > Thanks a lot for your response. After reading the response, I keep my rating since the novelty of this paper is not enough for ICLR.
> >
> > About response 1, in fact, authors state that the work is “first to explore the potential of multimodal foundation models for WSTVG”. In fact, the multimodal foundation models have been used in many multi-modal downstream tasks. I do not think that using the multimodal foundation models in a very small topic is novel enough for ICLR.
> >
> > About response 2: "To the best of our knowledge, our work is the first to study curriculum-based strategy for the WSTVG task". In fact, the curriculum-based strategy is very common in many related tasks, e.g., [1,2]. The difficulty measurer has been addressed in [2,3].
> >
> > About response 3: I do not think that, in CRG module, dividing a sentence into noun, adj and verb is novel and not explored.
> >
> > [1] Towards Robust Temporal Activity Localization Learning with Noisy Labels, COLING 2024
> >
> > [2] CCML: Curriculum and Contrastive Learning Enhanced Meta-Learner for Personalized Spatial Trajectory Prediction, TPAMI 2024
> >
> > [2] Answering from Sure to Uncertain: Uncertainty-Aware Curriculum Learning for Video Question Answering, Arxiv 2024

---

> > > ### Author Response · Authors · 2024-12-03
> > >
> > > We thank the reviewer for their time and for providing another opportunity to address the raised concern. We are sorry that the previous response did not adequately answer the questions. Here we will try our best to answer the question.
> > >
> > >
> > > **W.1. A very small topic, the novelty for ICLR, and the use of foundation models for STVG.**
> > >
> > > We respectfully but strongly disagree with the reviewer’s assessment regarding novelty and the perceived relevance of the task. Below, we address the specific points raised and provide clarifications:
> > >
> > > - **Very small topic Critique** - The suggestion that STVG is a “very small topic” is both factually and logically flawed. By this reasoning, a significant portion of recent work, including those leveraging foundation models for other tasks, would also fall under this classification. For instance:
> > > - ICLR 2024 works [1], [2] explore single tasks using foundation models.
> > > - CVPR 2024 [3] addresses STVG in a fully supervised setting via foundation model.
> > >
> > > STVG is far from trivial. It is, in fact, one of the most **challenging problem statements** in vision-language research. Performance improvement on VidSTG-Declarative and VidSTG-Interrogative datasets has been less than **2% over two years**, despite the emergence of multiple powerful foundation models. If STVG were as trivial as implied, it would have been “solved” by now. Instead, the persistent gap in progress underscores the complexity and importance of this task.
> > >
> > > - **Novelty** - The adaptation of foundation models for spatio-temporal video grounding (STVG) is, to the best of our knowledge, novel. This work represents a pioneering effort in this domain, introducing significant contributions such as:
> > >
> > > - Contextual Referral Grounding
> > > - Self-paced Scene Understanding
> > >
> > > If the reviewer still considers this work insufficiently novel for ICLR, we request concrete references to prior work addressing the STVG task using foundation models in weakly supervised settings. The absence of such citations would further support the novelty claim of this submission.
> > >
> > > - **Use of foundation model** - We thoroughly analyzed 60 dense multimodal foundation models and found that none addressed the STVG task. To ensure rigor, we benchmarked the Top-2 multimodal models for Image Referral tasks, specifically:
> > >
> > > - GroundingDINO (ECCV’24)
> > > - Lenna: Language Enhanced Reasoning Detection Assistant (ArXiv’23)
> > >
> > > GroundingDINO outperformed Lenna, leading us to adopt it as our feature extractor. However, trivial extensions of these models failed to generalize across datasets, further demonstrating the necessity of CoSPaL. The adoption of a multimodal foundation model for STVG is both well-justified and a significant step forward for conferences like ICLR.
> > >
> > > **W.2. Use of curriculum learning for WSTVG task.**
> > >
> > > We appreciate the reviewer’s effort in pointing out curriculum learning papers for comparison. However, we must strongly clarify that the references provided are misaligned with the scope of our work.
> > >
> > > The reviewer appears to be conflating **dense video tasks** with **global-level video tasks**. None of the referenced papers address the unique challenges of dense video tasks, let alone spatio-temporal video grounding (STVG). As stated in our previous response, methodologies developed for global-level video tasks are fundamentally inapplicable to STVG due to the significant differences in task granularity and requirements.
> > >
> > > Moreover, none of the referenced works address the problem in **weakly supervised settings**, where no ground truth annotations are available. This is a key aspect of our work and a major differentiator from existing curriculum learning literature.
> > >
> > > While we appreciate the attempt to provide additional references, we are confident in our ability to rebut comparisons and clearly demonstrate the distinctiveness and impact of our contribution. To avoid further confusion, we urge the reviewer to provide examples of works specifically addressing **dense video tasks** in **weakly supervised settings**, if such papers exist. Otherwise, this line of critique appears misplaced.
> > >
> > > **W.3. Novelty of CRG.**
> > >
> > > We want to strongly emphasize that this work is far from an incremental or marginal contribution, as the reviewer seems to imply. While we acknowledge the value of using POS tags in multimodal works, the critique appears misplaced in the context of referral grounding tasks.
> > >
> > > Referral grounding in existing tasks involves **noise-free textual queries**, a stark contrast to the queries in STVG. STVG queries contain substantial **background information and irrelevant context**, which significantly challenges the model’s ability to focus on the target subject, especially in the **absence of ground truth supervision**. This fundamental distinction underscores the inadequacy of directly applying existing methodologies to STVG.

---

> ### Author Response · Authors · 2024-11-21
>
> **W.3 Response continued...**
>
> CRG extracts contextual information from textual queries to enhance the model's attention and strengthen the model's spatio-temporal grounding capability (shown in Table 5 main paper). SPS looks into the limitations of TPG on its complex scene understanding. For this aspect, we propose an increasing scene complexity-based training paradigm to gradually increase the hardness of the task and improve the model's spatio-temporal grounding ability (shown in Table 5 main paper).
>
> **W.4 Main ablation study should be in the main paper.**
>
> Thanks for this comment. We will update the manuscript to show the main ablation study tables in the main paper.
>
> **W.5 Start and end timestamps - Fig. 4.**
>
> In Fig. 4, darker shade represents temporal boundaries, i.e. start and end timestamps. We provide this detail in the caption.
>
> **W.6 Typo - Writing.**
>
> Thanks for pointing it out. We will update the manuscript and correct the typo.
>
> [1] Weakly-Supervised Spatio-Temporal Video Grounding with Variational Cross-Modal Alignment, ECCV 2024.
>
> [2] WINNER: Weakly-supervised hIerarchical decompositioN and aligNment for spatio-tEmporal video gRounding, CVPR 2023.

---

> ### Author Response · Authors · 2024-11-24
>
> Dear Reviewer tfEJ,
>
> We are sincerely thankful for the time and work you put into reviewing our paper. We hope our answers clarify your queries and if you have any more queries regarding the paper feel free to ask us any time. We will be glad to answer them.
>
> Sincerely,
>
> Authors of Paper 1331.

---

> ### Author Response · Authors · 2024-11-30
>
> Dear Reviewer tfEJ,
>
> Once again, thanks for your comments. As the discussion period winds down soon, please follow up if our rebuttal clarified and answered your questions, and if we can answer or clarify additional points.
>
> Best,
>
> Authors of paper 1331.

---

> ### Author Response · Authors · 2024-12-03
>
> **W.3 Response continued...**
>
> The need to make the model **context-aware** is not merely an enhancement—it is essential for tackling the unique challenges of STVG. Our Contextual Referral Grounding (CRG) approach directly addresses this gap, and its effectiveness is clearly demonstrated in Table 5 of the main paper. While CRG may appear conceptually **simple**, its impact is both substantial and non-trivial, as evidenced by the significant performance improvements achieved.
>
> To dismiss this as incremental disregards both the complexity of the problem and the demonstrated novelty of our solution. We respectfully urge the reviewer to reassess this critique in light of the clear contributions and the task-specific challenges addressed in this work.
>
>
> **Conclusion**
>
> We firmly believe this work is novel, impactful, and timely for the research community. The reviewer’s critique lacks sufficient grounding in existing literature and context. If STVG or our approach is considered insufficient, we urge the reviewer to provide specific references to comparable or superior work. Without such substantiation, dismissing this effort undermines the principles of fair and constructive review.
>
> [1] VILMA: A ZERO-SHOT BENCHMARK FOR LINGUISTIC AND TEMPORAL GROUNDING IN VIDEO-LANGUAGE MODELS, ICLR 2024.
>
> [2] FERRET: REFER AND GROUND ANYTHING ANYWHERE AT ANY GRANULARITY, ICLR2024.
>
> [3] Video GroundingDINO - Towards Open-Vocabulary Spatio-Temporal Video Grounding, CVPR 2024.

---

### Official Review · Reviewer_4gRV · 2024-10-31

**Soundness:** 3
**Presentation:** 3
**Contribution:** 3
**Rating:** 6
**Confidence:** 4

**Summary:**

This paper proposes CoSPaL (Contextual Self-Paced Learning) for Weakly Supervised Spatio-Temporal Video Grounding (WSTVG). The method introduces three key components: Tubelet Phrase Grounding (TPG), Contextual Referral Grounding (CRG), and Self-Paced Scene Understanding (SPS). CoSPaL aims to improve spatio-temporal video grounding by enhancing the model’s ability to understand complex queries and progressively adapt to more difficult scenarios. The effectiveness of CoSPaL is validated on three benchmark datasets, with significant performance improvements over baselines.

**Strengths:**

1.	The paper presents a method CoSPaL to address the key issues in WSTVG.
2.	The paper includes comprehensive quantitative analysis and visualization, providing empirical evidence to support the proposed method.
3.	The proposed CoSPaL model shows strong performance gains on multiple datasets, with notable improvements of 3.9% on VidSTG and 7.9% on HCSTVG-v1.

**Weaknesses:**

1.	The technical contribution and motivation is unclear. For example, TPG module uses cross-modal attention, contrastive learning, and feature reconstruction to facilitate interactions between words and tubelets, these techniques are common in the field. Why this particular implementation contributes to improved spatial and temporal grounding?
2.	The SPS component is not clearly described. It is unclear how sample complexity is determined within the curriculum learning framework, is the object number? But intuitively, videos with more objects, faster changes, and more interactions are more complex.
3.	What does “Self-Paced” mean? This concept is not clearly explained in the paper.
4.	In Table 5, the introduction of SPS in the TPG module leads to a performance drop in vIoU@0.3 (TPG+SPS vs TPG). This raises concerns about the efficacy of SPS in certain settings. The authors should provide a deeper analysis of why this occurs.
5.	In page 10, the part of Impact on actor localization, the corresponding quantitative results for this claim are missing.

**Questions:**

See above

---

> ### Author Response · Authors · 2024-11-21
>
> We sincerely thank the reviewer for the valuable feedback and analysis of the paper. We have addressed the questions and concerns raised by the reviewer.
>
> **W.1 Technical contribution and motivations of TPG module.**
>
> Our contribution is an effective adaptation of multimodal foundation models for this task, which to the best of our knowledge has not been explored earlier. We agree with the reviewer that cross-modal attention, contrastive learning, and feature reconstruction are existing components, but we merely use these to develop a strong baseline since a trivial adaptation was not found effective (shown in Table 1 main paper). These existing components are carefully utilized considering the limitations of the trivial extension. The proposed method CoSPaL uses this adaptation as a baseline.
>
> CoSPaL consists of novel components: 1) Contextual referral Grounding (CRG) and 2) Self-paced Scene understanding (SPS) modules which to the best of our knowledge have not been explored earlier. CRG extracts contextual information from textual queries to enhance the model's attention and strengthen the model's spatio-temporal grounding capability (shown in Table 5 main paper). SPS looks into the limitations of TPG on its complex scene understanding. For this aspect, we propose an increasing scene complexity-based training paradigm to gradually increase the hardness of the task and improve the model's spatio-temporal grounding ability (shown in Table 5 main paper).
>
> **W.2 Clarification on SPS component.**
>
> The sample complexity is defined based on spatio-temporal scene complexity with a maximum number of tubelets failing with TPG (baseline adapted from Grounding DINO). It is not only based on the number of objects. To prove this hypothesis, we look into the failure cases for the number of tubelets ranging from 2 to 10. We observe that the maximum number of failure percentage is with four and seven tubelets. Ideally, it should be at the far end 8-10 tubelets. This is attributed to multiple reasons for spatio-temporal tasks such as challenging poses, scene changes (high displacement), partial body availability, etc. We agree with the reviewer that the number of tubelets might be intuitive. However, we deduce the scene complexity via baseline failure rate. To the best of our knowledge, our work is the first to address this for the WSTVG task. We propose an empirical approach to determine schedule stages suitable for the WSTVG task, and we also demonstrate that this is generalizable to other datasets.
>
> **W.3 Clarification on self-paced.**
>
> In our context, self-paced means the hardness of a task where the difficulty of training is controlled from a student’s perspective as to how to increase the complexity of a scene gradually.
>
> **W.4 Performance drop at vIoU@0.3 (Table 5).**
>
> Thanks for pointing it out. It’s a minor typo. The performance at vIoU@0.3 is 30.47 for TPG+SPS. We will update the manuscript and fix the typo.
>
> **W.5 Impact on actor localization Table.**
>
> We provide the table here *Table R31*. The table is presented in the supplementary (Table 12 a). We will update the manuscript to include the table in the main paper.
>
> | Method | Accuracy |
> |-|-|
> | W-GDINO | 18.7 |
> | TPG | 35.5 |
> | CoSPAL | 38.7 |
>
> *Table R31: Ablation on classification accuracy spatial actor localization.*

---

> ### Author Response · Authors · 2024-11-24
>
> Dear Reviewer 4gRV,
>
> We are sincerely thankful for the time and work you put into reviewing our paper. We hope our answers clarify your queries and if you have any more queries regarding the paper feel free to ask us any time. We will be glad to answer them.
>
> Sincerely,
>
> Authors of Paper 1331.

---

> > ### Comment · Reviewer_4gRV · 2024-11-27
> >
> > Thanks for the authors' feedback. Most of my concerns have been addressed. I decide to keep my positive rating.

---

### Official Review · Reviewer_BmrP · 2024-11-01

**Soundness:** 2
**Presentation:** 3
**Contribution:** 3
**Rating:** 5
**Confidence:** 4

**Summary:**

Targeting the issues of inconsistent predictions in time judgment, difficulty in comprehending complex queries, and the complexity of application scenarios faced by weakly supervised Spatio-Temporal Video Grounding (STVG), this paper introduces the self-paced learning method, which has achieved certain performance improvements on two conventional datasets.

**Strengths:**

1. The problem addressed in this paper is of certain significance and is interesting.
2. The introduction of self-paced learning is reasonably justified.
3. Experimental results on different datasets indicate that the algorithm has achieved certain improvements.

**Weaknesses:**

1. The motivation of introducing self-paced learning should be illustrated more clearly. Are there other methods that can address the challenges presented in this paper? It is suggested to provide a more detailed explanation in the related work section.
2. In Section 3.2.1, the tracking algorithm in the TPG module seems to play a vital role in learning the whole module. It makes a good start with GroundingDINO in the whole model, and it would be beneficial to add an analysis of the ablation of the tracking algorithm.
3. Also, in section 3.2.1, the part of the Spatial Grounding Module, the formula description seems confused. It does not give key dimension information, such as in the similarity calculations: $\text{SIM}(f_{w_m},f_{\tilde{T_k}})=(\mathrm{MLP_{q}}(f_{w_{m}})^{T}\mathrm{MLP_{k}}(f_{\tilde{T_{k}}}))/\sqrt{d}$, where $f_{w_m}\in \mathbb{R}^{1\times768}$ and $f_{\tilde{T_{k}}}\in\mathbb{R}^{T\times{256}}$ obtains from the above, so the $MLP_q(f_{w_m})\in\mathbb{R}^{1\times{D}}, MLP_{k}(f_{\tilde{T_{k}}})\in\mathbb{R}^{T\times{D}}$, where $D$ is the MLP output dimension, but if it is like this, then matrix multiplication is failed. Besides, in $\mathrm{A_T}(f_{T}, f_{w_{m}})=\sum_{k=1}^{K}\operatorname{softmax}\left(f_{\tilde{T}_{k}}, f_{w_{m}}\right) \mathrm{MLP}_{v}\left(f_{\tilde{T}_{k}}\right)$, what is the mean about $f_T$? Is $\sum_{k=1}^{K}$ meant to be a matrix addition of all tubelet features? Then it will give $ A_\in\mathbb {R}^{1\times{D}}$; at this point, how to get the distribution of scores between different tubelets? This part of the equation is confusing.
4. The comparison parameters of GPU memory use and training time in Figure 5 are ambiguous because the fully supervised models compared in the figure are experimented on different resolutions. However, the paper does not list the information about the resolution of the fully supervised models compared at the time of statistics and the resolution of their own models and whether the hardware parameters (e.g., information about CPU, GPU, memory) are unified across the different models, which I think are essential settings.

**Questions:**

Please refer to the Weaknesses.

---

> ### Author Response · Authors · 2024-11-21
>
> We sincerely thank the reviewer for the valuable feedback and analysis of the paper. We have addressed the questions and concerns raised by the reviewer.
>
> **W.1 Motivation of self-paced learning. Other methods that can address this challenge.**
>
> - The motivation of SPS is to increase the spatio-temporal scene complexity gradually for better optimization of the model. We first analyze the limitations of TPG qualitatively (fig 2 c main paper) and quantitatively. Quantitatively, analyzing TPG (adaptation of foundation model), we analyze the scenes based on spatio-temporal complexity and observe the maximum failure rate of baseline adaptation, TPG. We found out that failure cases are specifically higher for scenes with four and seven tubelets. We propose an empirical approach to determine schedule stages suitable for the WSTVG task, and we also demonstrate that this is generalizable to other datasets.
>
> - We look into other methods which address the challenge via curriculum learning. These approaches can’t be directly extended to STVG task, mostly for one significant reason: Difficulty is defined based on whether images/regions (image classification, object detection, segmentation, etc.) are difficult to classify from a fixed set of classes whereas STVG task is a free-form textual query understanding. STVG adds an additional level of complexity via spatio-temporal tubelets. To the best of our knowledge, no work addresses the self-paced curriculum learning strategy for this scenario.
>
> - We will include a section in related works discussing previous works on curriculum learning.
>
> **W.2 Ablation study on tracker algorithm.**
>
> For the rebuttal, we ran an additional experiment on two different state-of-the-art tracking algorithms, StrongSORT and Byte Tracker on the HCSTVG-v1 dataset. We ran an upper-bound analysis on both algorithms. We share the performance in Table *R 21*. Comparing the tracker used in our work (BoTSort), BotSort outperforms ByteTrack. The performance difference between BotSort and StrongSort is <1%. However, using a better tracker algorithm will boost our performance further. We will include this ablation study in the revised manuscript.
>
> | Tracker | tIoU | vIoU | vIoU@0.3 | vIoU@0.5 |
> |-|-|-|-|-|
> | BotSort (Ours) | 79.23 | 62.28 | 84.73 | 69.54 |
> | StrongSort | 80.31 | 62.89 | 85.42 | 71.70 |
> | ByteTrack | 77.00 | 60.06 | 82.14 | 66.70 |
>
> *Table R21: Ablation study on upper-bound analysis with different tracking algorithms.*
>
> **W.3 Clarification on spatial grounding module equation.**
>
> Thanks for this comment. In the SIM equation, we calculate the dot product per frame and then take the mean across the frame dimension to aggregate the attention across the tubelet. For the next equation clarification, we sample all the tubelets present in a video in the pre-processing stage. We calculate the attention between words and each tubelet’s frame features and then accumulate the attention score and take the mean over it.
>
> **W.4 Comparison between GPU memory.**
>
> CG-STVG[1] paper provides an analysis of computation comparison on recent fully-supervised approaches (Table-2 supplementary). We utilize the information for fully-supervised approaches from CG-STVG. We share the details in Table *R32*. The code for CSDVL is not provided. We will revise the manuscript to include this table.
>
> | **Methods**   | **Input Resolution** |**Trainable Params**|  **Total Params** | **Training Time** | **GPU Mem (GB)** | **GPU Num** |  **FLOPs** |
> |---------------|----------------------|------------------|------------------|--------------------|------------------|-------------|-----------|
> | TubeDETR [2]  | 352 | 185 |  185              | 48 h              | 29.9             | 16 V100     | 1.45 T    |
> | STCAT [3]     | 448  | 207 |  207              | 12 h              | 39.2             | 32 A100     | 2.85 T    |
> | CSDVL [4]     | -  | - | -                | ~48 h             | -                | 8 A6000     | -         |
> | CG-STVG [1]       | 420 | 203 |  231              | 13.6 h            | 43.9             | 32 A100     | 3.03 T    |
>
> *Table R32: Ablation Table on the computation of fully-supervised approaches.*
>
> [1] Gu, Xin, et al. "Context-Guided Spatio-Temporal Video Grounding." Proceedings of the IEEE/CVF Conference on Computer Vision and Pattern Recognition. 2024.
>
> [2] Yang, Antoine, et al. "Tubedetr: Spatio-temporal video grounding with transformers." Proceedings of the IEEE/CVF Conference on Computer Vision and Pattern Recognition. 2022.
>
> [3] Jin, Yang, Zehuan Yuan, and Yadong Mu. "Embracing consistency: A one-stage approach for spatio-temporal video grounding." Advances in Neural Information Processing Systems 35 (2022): 29192-29204.
>
> [4] Z. Lin, C. Tan, J. -F. Hu, Z. Jin, T. Ye and W. -S. Zheng, "Collaborative Static and Dynamic Vision-Language Streams for Spatio-Temporal Video Grounding," 2023 IEEE/CVF Conference on Computer Vision and Pattern Recognition (CVPR), Vancouver, BC, Canada, 2023.

---

> > ### Comment · Reviewer_BmrP · 2024-11-26
> > **Reply**
> >
> > Thank you for the response, which resolved some of my concerns. However, I still have some confusion. Given that Grounding DINO already obtains a bounding box, BoTSORT seems to primarily play a refining role. For Q2, what I am actually interested in is the ablation study on BoTSORT, as this would demonstrate the novelty of the entire workflow. Additionally, due to the complexity of the process, could the authors provide an anonymous codebase to better understand this work?

---

> > > ### Author Response · Authors · 2024-11-27
> > >
> > > We thank the reviewer for their time and for providing another opportunity to address the raised concern. We are sorry that the previous response did not adequately answer the questions. Here we will try our best to answer the question.
> > >
> > > **W.1. Ablation on BoTSort.**
> > >
> > > There is a little confusion about what details are required. We interpret the query in three possible ways and address all aspects below. If the reviewer seeks clarification on additional details, we would be happy to elaborate further.
> > >
> > > - **Use of BoTSort as post-processing step** - BoTSort serves as a post-processing step in our method to link bounding boxes across frames, enabling tracking of subjects as they enter and leave the scene throughout the video. This approach aligns with earlier works, such as [1], where tracker algorithms were similarly employed as post-processing steps to temporally link bounding boxes.
> > >
> > > - **BoTSort hyperparameters** - BotSort depends on a few hyperparameters, including lower and upper confidence thresholds, thresholds for initializing new tracks, and spatial IoU thresholds for linking tracks. For the rebuttal, we conducted additional experiments by tuning these hyperparameters. The results reveal that variations in the mean vIoU remain under 1%, suggesting that our method's performance is robust to these parameters.
> > >
> > > - **Inference at test Time - Tubelet vs Individual Frame** - For the rebuttal, to further analyze the role of tubelets in our approach, we performed additional experiments comparing tubelet-based grounding with frame-level grounding at inference time. The results, presented in Table *R33*, indicate that tubelets significantly enhance performance on the STVG task. Nonetheless, our approach outperforms previous methods even when grounding is performed at the individual frame level. We will incorporate this analysis and the associated table into the manuscript to provide additional clarity.
> > >
> > > | Approach | Frame | Tubelet | mean vIoU |  vIoU@0.3 | vIoU@0.5 |
> > > |-|-|-|-|-|-|
> > > | TPG | Yes | | 18.2 | 19.5 | 6.3 |
> > > | TPG | | Yes | 19.2 | 28.8 | 15.3 |
> > > | CoSPal | Yes | | 20.4 | 23.7 | 7.7 |
> > > | CoSPal | | Yes | 22.1 | 31.8 | 19.6 |
> > >
> > > *Table R33: Ablation on frame vs Tubelets on TPG and CoSPaL.*
> > >
> > > **W2: Code Access**
> > >
> > > The code has been uploaded as part of the supplementary materials. For reproducibility purposes, we have included the main training, evaluation, and model files while omitting specific sections of code. If the reviewer requires any further clarification or additional components in the codebase, we will be happy to assist.
> > >
> > > We hope our responses adequately address your concerns. If there are any further queries or additional clarifications required, we would be glad to provide them. Thank you once again for your valuable feedback and insights.
> > >
> > > [1] WINNER: Weakly-supervised hIerarchical decompositioN and aligNment for spatio-tEmporal video gRounding, CVPR 2023

---

> ### Author Response · Authors · 2024-11-24
>
> Dear Reviewer BmrP,
>
> We are sincerely thankful for the time and work you put into reviewing our paper. We hope our answers clarify your queries and if you have any more queries regarding the paper feel free to ask us any time. We will be glad to answer them.
>
> Sincerely,
>
> Authors of Paper 1331.

---

> ### Author Response · Authors · 2024-11-30
>
> Dear Reviewer BmrP,
>
> Once again, thanks for your comments. As the discussion period winds down soon, please follow up if our rebuttal clarified and answered your questions, and if we can answer or clarify additional points.
>
> Best,
>
> Authors of paper 1331.

---

### Official Review · Reviewer_qQgs · 2024-11-03

**Soundness:** 3
**Presentation:** 3
**Contribution:** 3
**Rating:** 8
**Confidence:** 4

**Summary:**

This paper introduces CoSPaL (Contextual Self-Paced Learning), a novel approach to Weakly Supervised Spatio-Temporal Video Grounding (WSTVG), which aims to locate objects in videos based on text descriptions without requiring detailed bounding box annotations during training. CoSPaL proposes three main components to overcome limitations of existing methods: Tubelet Phrase Grounding (TPG) for improved object tracking, Contextual Referral Grounding (CRG) for better query comprehension, and Self-Paced Scene Understanding (SPS) for progressive learning of complex scenarios. Results are reported on common grounding benchmarks such as VidSTG and HC-STVG-v1/v2.

**Strengths:**

1. The idea is innovative and well motivated.
2. The paper is well-written and easy to follow.
3. The method achieves strong performance improvements with respect to the baselines.

**Weaknesses:**

I have a few major concerns regarding the proposed method:

1. Regarding the Video Encoder: Existing works such as STCAT and TubeDETR utilize a Resnet-101 backbone to encode each frame. Even recent works such as VGDINO use the Grounding-DINO frozen image encoder. With this method using I3D as a separate video encoder, it could be an unfair comparison with existing works. How does this method perform if the authors utilize the same features as produced by the Grounding-DINO backbone as the video features? To make it consistent with previous works.

2. The method uses a pretrained tracker in the pipeline. I am concerned regarding the impact of this tracker, and how the performance changes if a different tracker is used? There does not seem to be any ablations regarding this.

**Questions:**

Please consider the weaknesses section. Essentially, my concerns are regarding the impact of the video encoder and the impact of the choice of tracker. I wish to understand how sensitive the method is to these two components, and whether the improvements are coming from the overall proposed design or certain specific components. I will further discuss this with fellow reviewers before making a final decision on the rating.

---

> ### Author Response · Authors · 2024-11-21
>
> We sincerely thank the reviewer for the valuable feedback and analysis of the paper. We have addressed the questions and concerns raised by the reviewer.
>
> **W.1 Use of Grounding DINO frame features.**
>
> We are currently running the experiments. We will update the results in the next 1-2 days.
>
> **W.2 Ablation study on Tracker algorithms.**
>
> For the rebuttal, we ran an additional experiment on two different state-of-the-art tracking algorithms, StrongSORT and Byte Tracker on the HCSTVG-v1 dataset. We ran an upper-bound analysis on both algorithms. We share the performance in Table *R 11*. Comparing the tracker used in our work (BoTSort), BotSort outperforms ByteTrack. The performance difference between BotSort and StrongSort is <1%. However, using a better tracker algorithm will boost our performance further. We will include this ablation study in the revised manuscript.
>
> | Tracker | tIoU | vIoU | vIoU@0.3 | vIoU@0.5 |
> |-|-|-|-|-|
> | BotSort (Ours) | 79.23 | 62.28 | 84.73 | 69.54 |
> | StrongSort | 80.31 | 62.89 | 85.42 | 71.70 |
> | ByteTrack | 77.00 | 60.06 | 82.14 | 66.70 |
>
> *Table R11: Ablation study on upper-bound analysis with different tracking algorithms.*

---

> ### Author Response · Authors · 2024-11-24
>
> Dear Reviewer qQgs,
>
> We are sincerely thankful for the time and work you put into reviewing our paper. We hope our answers clarify your queries and if you have any more queries regarding the paper feel free to ask us any time. We will be glad to answer them.
>
> Sincerely,
>
> Authors of Paper 1331.

---

> > ### Comment · Reviewer_qQgs · 2024-11-26
> >
> > Dear Authors,
> >
> > My concern regarding the tracking algorithm is now clear and I thank you for providing the additional ablation. However, I am still waiting for your response regarding the grounding DINO frame features. You mentioned that it would be done in a couple of days. Is there any update on that? Because that defines the difference between improvements coming from your method or just the usage of a different backbone.

---

> > > ### Author Response · Authors · 2024-11-26
> > >
> > > We have the numbers using gdino image encoder features for the TPG module.
> > >
> > >
> > > | Encoder | tIoU | vIoU | vIoU@0.3 | vIoU@0.5 |
> > > |-|-|-|-|-|
> > > | I3D (Ours) | 37.6| 19.2| 28.8| 15.3|
> > > | GDino  | 37.0| 19.4| 29.4| 14.7|
> > >
> > > We observe the performance is almost similar with different backbone features. This signify that our approach is independent of backbone.

---

> > > > ### Comment · Reviewer_qQgs · 2024-11-27
> > > >
> > > > Dear Authors,
> > > >
> > > > Thank you for your response. All my concerns are addressed. I have increased my rating.

---

> > > > > ### Author Response · Authors · 2024-12-02
> > > > >
> > > > > Dear Reviewer qQgs,
> > > > >
> > > > > Thank you very much for your thoughtful reconsideration and for raising the rating. We deeply appreciate your valuable feedback and will certainly take all the reviews into account as we improve the final version of the paper.
> > > > >
> > > > > Thank you again for your support!
> > > > >
> > > > > Best regards,
> > > > >
> > > > > Authors

---

### Author Response · Authors · 2024-11-21

We sincerely thank all the reviewers for the valuable feedback and analysis of the paper. The reviewers acknowledged the following strengths:

1. **Stronger** performance across multiple datasets. (Reviewers: All)
2. Motivation is **well articulated** and the **good flow** of the paper. (Reviewers - qQgs, BmrP)
3. **Easy** to follow up on the proposed approach with **good tables/figures**. (Reviewers - qQgs, tfEJ)
4. **Comprehensive** analysis. (Reviewers - 4gRV, tfEJ)

Here, we would like mention the motivation and novelty of our proposed work.

**Motivations of TPG module.**

TPG is an effective adaptation of multimodal foundation models for this task, which to the best of our knowledge has not been explored earlier. We agree with the reviewers that cross-modal attention, contrastive learning, and feature reconstruction are existing components, but we merely use these to develop a strong baseline since a trivial adaptation was not found effective (shown in Table 1 main paper). These existing components are carefully utilized considering the limitations of the trivial extension. The proposed method CoSPaL uses this adaptation as a baseline.

**Technical contributions of CoSPaL.**

CoSPaL consists of novel components: 1) Contextual referral Grounding (CRG) and 2) Self-paced Scene understanding (SPS) modules which to the best of our knowledge have not been explored earlier. CRG extracts contextual information from textual queries to enhance the model's attention and strengthen the model's spatio-temporal grounding capability (shown in Table 5 main paper). SPS looks into the limitations of TPG on its complex scene understanding. For this aspect, we propose an increasing scene complexity-based training paradigm to gradually increase the hardness of the task and improve the model's spatio-temporal grounding ability (shown in Table 5 main paper).

**Motivation of self-paced learning. Other methods that can address this challenge.**

- The motivation of SPS is to increase the spatio-temporal scene complexity gradually for better optimization of the model. We first analyze the limitations of TPG qualitatively (fig 2 c main paper) and quantitatively. Quantitatively, analyzing TPG (adaptation of foundation model), we analyze the scenes based on spatio-temporal complexity and observe the maximum failure rate of baseline adaptation, TPG. We found out that failure cases are specifically higher for scenes with four and seven tubelets. We propose an empirical approach to determine schedule stages suitable for the WSTVG task, and we also demonstrate that this is generalizable to other datasets.

- We look into other methods which address the challenge via curriculum learning. These approaches can’t be directly extended to STVG task, mostly for one significant reason: Difficulty is defined based on whether images/regions (image classification, object detection, segmentation, etc.) are difficult to classify from a fixed set of classes whereas STVG task is a free-form textual query understanding. STVG adds an additional level of complexity via spatio-temporal tubelets. To the best of our knowledge, no work addresses the self-paced curriculum learning strategy for this scenario.

---

### Meta-Review · Area_Chair_8dbS · 2024-12-20

**Metareview:**

The paper proposes CoSPaL, in approach for Weakly Supervised Spatio-Temporal Video Grounding (WSTVG) by leveraging foundational models (i.e., Grounding DINO). Due to the strong improvement brought by CoSPaL and solid ablation experiments, two reviewers support to accept the paper (qQgs and 4gRV) while two other reviewers (BmrP, tfEJ ) still have concerns about the paper: novelty and missing ablations comparisons.

**Additional Comments On Reviewer Discussion:**

Below are discussions happening between reviewers and authors:

**For reviewer BmrP**: The reviewer asks about further ablations on (tracker algorithm, BoTSort) and GPU memory comparison, which the authors further provide ablation results which show positive signal for CoSPaL. The reviewer furthers asks about code release in which the authors had provided in their supplementary.

**For reviewer tfEJ**: The reviewer mainly concerns about the novelty which basically argues that CoSPaL is leveraging foundational models for small tasks, here is WSTVG. In the rebuttal, the authors clearly state that the novelty of CoSPaL consists of novel components: Contextual referral Grounding (CRG) and Self-paced Scene understanding (SPS) and clarify why it is non-trivial to adapt foundational models to WSTVG. Many discussions are on clarifying the curriculum learning is applied on dense prediction task as opposed to global tasks.

The area chair reads all reviews and discussions and agrees with the authors that it is not trivial to adapt foundational models to WSTVG. The area chair finds the improvements and solid ablations out-weight the novelty concern, plus the authors will release the code to further facilitate reproducibility, thus recommends to accept this paper.

---

### Decision · Program_Chairs · 2025-01-22

Accept (Poster)